# The interplay between the polar growth determinant DivIVA, the segregation protein ParA, and their novel interaction partner PapM controls the *Mycobacterium smegmatis* cell cycle by modulation of DivIVA subcellular distribution

Izabela Matusiak,[1] Agnieszka Strzałka,[1] Patrycja Wadach,[1] Martyna Gongerowska-Jac,[1] Ewa Szwajczak,[2] Aleksandra Szydłowska-Helbrych,[1] Bernhard Kepplinger,[1] Monika Pióro,[1] Dagmara Jakimowicz[1]

**ABSTRACT** Bacterial chromosome segregation is facilitated by the ParABS system. The ParB protein binds centromere-like *parS* sequences and forms nucleoprotein complexes. These nucleoprotein complexes are segregated by the dynamic ATPase ParA. In mycobacteria, ParA also interacts with the polar growth determinant DivIVA (Wag31). This interaction was earlier shown not only to facilitate the segregation of ParB complexes but also to affect cell extension. Here, we identify an additional partner of ParA in *Mycobacterium smegmatis*, named PapM. Using *E. coli*-based analysis, we show that PapM likewise interacts with DivIVA and that the tripartite interaction of ParA-PapM-DivIVA is phosphorylation dependent: ParA binding to DivIVA is diminished, while PapM binding is promoted upon phosphorylation of DivIVA. The presence of PapM enhances the dissociation of ParA from the DivIVA complex upon its phosphorylation. Studies of *M. smegmatis* mutant strains reveal that altered PapM levels influence chromosome segregation and cell length. The elimination of PapM affects ParA dynamics. Furthermore, ParA and, to a lesser extent, PapM modulate the subcellular distribution of DivIVA. Altogether, our studies show that the tripartite interplay between ParA-DivIVA and PapM controls the switch between cell division and cell elongation and, in this way, affects the mycobacterial cell cycle.

**IMPORTANCE** The genus of *Mycobacterium* includes important clinical pathogens (*M. tuberculosis*). Bacteria of this genus share the unusual features of their cell cycle such as asymmetric polar cell elongation and long generation time. Markedly, control of the mycobacterial cell cycle still remains not fully understood. The main cell growth determinant in mycobacteria is the essential protein DivIVA, which is also involved in cell division. DivIVA activity is controlled by phosphorylation, but the mechanism and significance of this process are unknown. Here, we show how the previously established protein interaction partner of DivIVA in mycobacteria, the segregation protein ParA, affects the DivIVA subcellular distribution. We also demonstrate the role of a newly identified *M. smegmatis* DivIVA and ParA interaction partner, a protein named PapM, and we establish how their interactions are modulated by phosphorylation. Demonstrating that the tripartite interplay affects the mycobacterial cell cycle contributes to the general understanding of mycobacterial growth regulation.

**KEYWORDS** chromosome segregation, mycobacteria, polar growth, ParA

Key processes of the cell cycle must be precisely coordinated. In bacteria, chromosome replication overlaps with chromosome segregation. Both processes are

Address correspondence to Dagmara Jakimowicz, dagmara.jakimowicz@uwr.edu.pl.

The authors declare no conflict of interest.

See the funding table on p. 19.

accompanied by cell elongation and must be completed before cell division (1–3). Most bacterial species employ a system based on ParA and ParB proteins to actively segregate their chromosomes (3–5). ParAB proteins move and re-position chromosomal replication initiation regions (*oriC*) soon after they are duplicated. ParB binds to *oriC* proximal *parS* sites to form nucleoprotein segregation complexes named segrosomes, which are moved toward cell poles due to interactions with the nucleoid-bound ATPase ParA (5, 6). Upon interaction with segrosomes, ParA dimers hydrolyze ATP and are released from the nucleoid, while the ParB complex moves toward a higher concentration of the nucleoid-bound ParA (7, 8). Surprisingly, the ParAB-mediated segregation shows some bacterial species-specific features (4, 9, 10). These differences involve the interactions of the segregation proteins with other proteins or protein complexes, like the cell elongation complex which governs polar cell extension in Actinobacteria, such as mycobacteria.

Mycobacteria are well known as slow-growing, clinically important pathogens (e.g., *Mycobacterium tuberculosis*), but this genus also contains relatively fast-growing saprophytes (e.g., *Mycobacterium smegmatis*, whose division time is approximately 150 min) (11, 12). Mycobacteria are rod-shaped bacilli, characterized by a thick, lipid-rich cell wall and uncommon features of the cell cycle (13–15). Their slow growth as well as the ability to remain in a dormant, nonreplicating state for years increase their potential of being successful intracellular pathogens (11). Interestingly, the key mechanisms of mycobacterial cell cycle control are still not well understood.

One of the unique mycobacterial features is their asymmetric polar cell extension: the cell pole inherited from the mother cell (the old cell pole) extends faster than the newly established cell pole (the new cell pole) (11, 16–19). The asymmetry of the mycobacterial cell is conferred by the polar protein complex that governs cell extension. The key component of this polar complex is the essential, polarity determination protein DivIVA (in mycobacteria also called Wag31 or Antigen 84) (20–23). Changes in the DivIVA level modulate polar cell wall synthesis and affect cell shape (20, 21, 23, 24). DivIVA was shown to control the activity of the enzymes involved in the cell wall synthesis (25, 26). At the latest stages of cell division, DivIVA is recruited to the division site where it interacts with divisome protein PBP3 (27, 28). It was also suggested that the other DivIVA interaction partners CwsA and CrgA may contribute to DivIVA relocation (29). The late recruitment of DivIVA at the cell division site suggests its involvement in constituting the new cell poles (11, 19). DivIVA mutational analysis suggested that DivIVA activity involves different interactions at the new pole, the old pole, and the septum (27). However, the details of the DivIVA role in the formation of the new poles are still to be investigated.

Mycobacterial DivIVA was shown to be phosphorylated at a single threonine (Thr73 in *M. tuberculosis*, Thr74 *M. smegmatis*) by PknA and PknB kinases in response to environmental conditions (20, 25, 30). DivIVA phosphorylation is further controlled by MtrB, which is the sensor kinase of the two-component system MtrAB (31). DivIVA phosphorylation reportedly increased its polar accumulation. The modification was elevated during the exponential growth phase and was linked to enhanced cell wall synthesis leading to faster cell extension (20, 25, 32). Thus, phosphorylation-dependent DivIVA activity was suggested to link the rate of the mycobacterial cell extension to environmental conditions. However, the exact mechanism of this process remains underexplored.

Markedly, the asymmetry of the mycobacterial cells is also manifested by the asymmetry of their cell division, chromosome positioning, and segregation. Mycobacteria, like other bacteria, use ParA and ParB proteins to segregate the newly replicated *oriC* regions (33, 34). While in *M. tuberculosis*, *parA* and *parB* genes are essential, in *M. smegmatis*, elimination of ParA or ParB is possible but leads to severe chromosome segregation defects and alters the cell length (34, 35). In *M. smegmatis*, ParB binds three *parS* sites located in the proximity of the *oriC* region to form a segrosome (35, 36). Upon initiation of the DNA replication, duplicated segrosomes move asymmetrically—the one that moves toward the new pole travels faster and covers a greater distance than the one that moves toward the old cell pole. The positioning and movements of the segrosomes

are governed by their interactions with the ParA protein (35). In *M. smegmatis*, EGFP-ParA is observed to accumulate transiently in proximity to the new cell pole in the new-born cell. During the progress of cell cycle, ParA-EGFP forms a fluorescent patch that extends along the cell, toward the old pole and ParB complex. In the absence of ParA, segrosomes are mislocalized, and cell division is often misplaced, resulting in the frequent formation of anucleate cells and the increased variation of the progeny cell length (35).

The unique features of mycobacterial chromosome segregation further include the interaction between ParA and DivIVA (9, 34, 37). Disruption of ParA-DivIVA binding results in modest segregation defects (37). However, the lack of ParA recruitment to DivIVA impairs segrosome movement and accelerates the cell elongation, implying that ParA affects DivIVA function. Interestingly, a point mutation that disrupts ParA binding to DNA leads to increased ParA accumulation at the cell poles, suggesting a competition between DivIVA and the nucleoid for ParA binding. Our studies have also suggested that polar recruitment of ParA may be induced by unfavorable environmental conditions (37).

Here, we further elucidate the mechanism of mycobacterial chromosome segregation. We identified, via a bacterial two hybrid screen, the novel component of *M. smegmatis* chromosome segregation machinery, protein MSMEG_5597 named PapM (ParA partner in *M. smegmatis*) that interacts with both ParA and DivIVA. We investigated how PapM modulates the interaction between ParA and DivIVA and how phosphorylation of the latter affects this interaction. We demonstrate that PapM cooperates with ParA in chromosome segregation and controls cell length. Finally, we explore PapM's influence on ParA dynamics and the impact of both partners on DivIVA subcellular distribution.

## RESULTS

### An interaction screen identified the novel ParA interaction partner named PapM

We sought to identify novel components of *M. smegmatis* chromosome segregation machinery. To this end, we screened the bacterial two-hybrid (BTH) library of *M. smegmatis* genome in the pKT25 vector using T18-ParA (pUT18C*parA*) as the bait. In the BTH system, the interaction between a bait and a pray, both fused to adenyl cyclase subunits (T18 and T25), restores adenyl cyclase activity in *E. coli* BTH101 cells and leads to development of colored colonies on an indicator media (38).

In the library screening among the positive clones, we identified one containing the fragment (encoding 8–206 aa) of the *msmeg_5597* gene hereafter named *papM* (ParA partner in *M. smegmatis*). The *papM* gene encoded for a 22.5-kDa protein consisting of 206 amino acids and belonging to a family of TetR regulators which are characterized by the presence of N-terminal HTH motif and ability to dimerize (39). To confirm the interaction of PapM with *M. smegmatis* ParA, we have cloned the entire *papM* gene into BTH vectors obtaining pUT18C*papM* and pKT25*papM*. BTH analysis confirmed the interaction between T25-ParA and T18-PapM, as well as between T25-PapM and T18-ParA (Fig. 1A; Fig. S1A). Additionally, we noted the interaction between T18-PapM and T25-PapM (Fig. 1A), which indicates the dimerization of PapM protein consistent with dimerization of TetR family regulators (39).

To confirm the observed interaction between ParA and PapM, we performed an affinity chromatography pulldown. In this approach, the glutathione-S-transferase (GST)-ParA fusion protein or the GST as the negative control was bound to GSH-Sepharose. Next, the lysate of *E. coli* BL21 overproducing His-PapM was applied to the column. Analysis of the affinity chromatography eluates using SDS-PAGE and Western blot with anti-His antibodies showed that His-PapM was retained on the resin in presence of GST-ParA but not in presence of GST alone (Fig. 1B). This experiment confirmed the direct interaction between PapM and ParA.

Having confirmed ParA-PapM interaction *in vitro*, we used BTH to examine if PapM preferably interacted with ParA dimer or monomer and if ParA's binding to DNA could influence its interaction with PapM. To this end, we tested the interaction of PapM

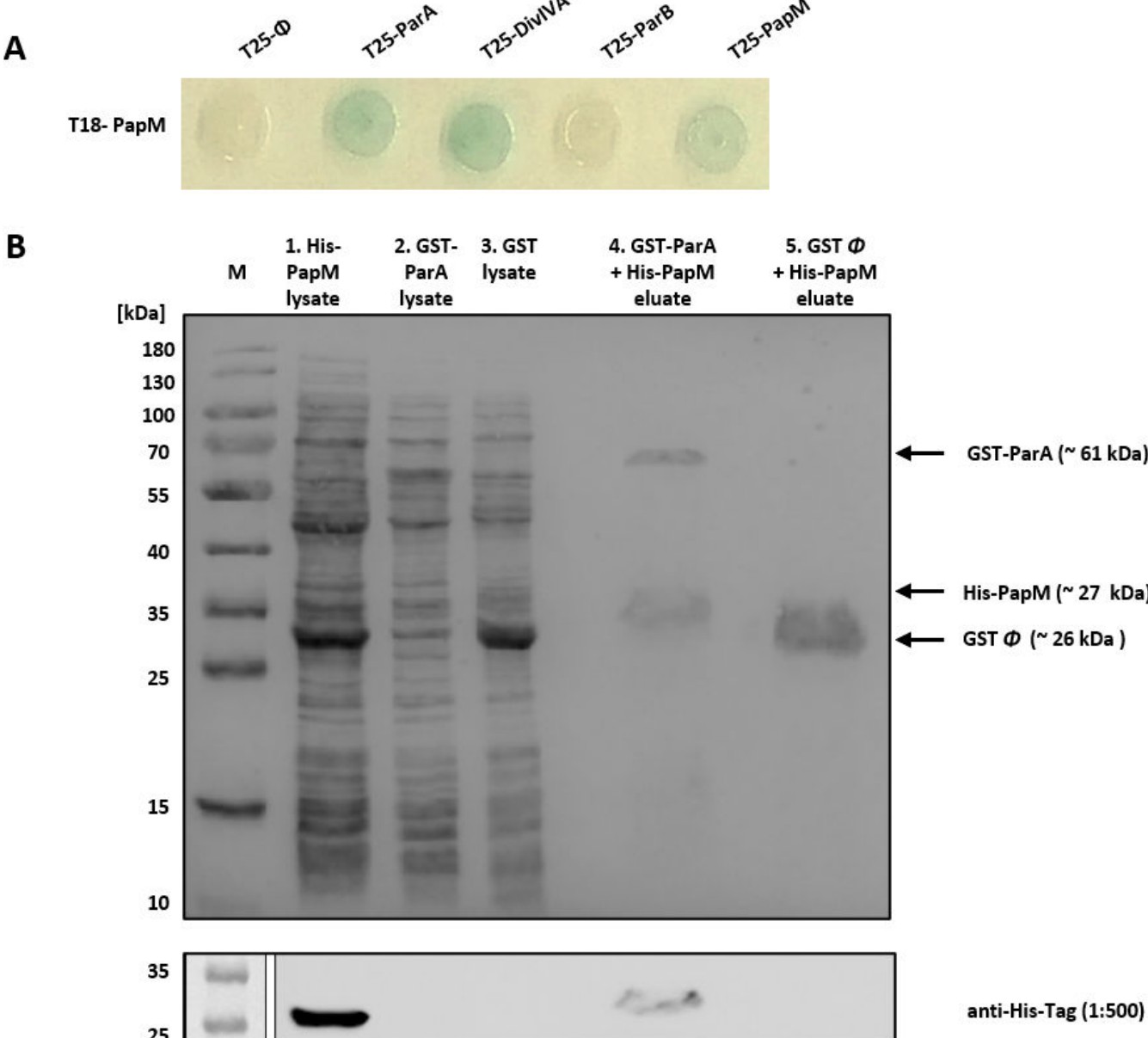

**FIG 1** Identification of PapM as a novel *M. smegmatis* ParA interaction partner. (A) BTH analysis shows T18-PapM interactions. The blue color of the colony indicates interaction. (B) Affinity chromatography confirmation of PapM-ParA interactions. The SDS-PAGE showing the His-PapM binding to GST-ParA but not to the GST column. First, the lysate of the *E. coli* BL21 producing GST-ParA (lane 2) or GST (lane 3) was loaded on GSH-Sepharose; next, after the washing of unbound proteins, lysate of the *E. coli* BL21 producing His-PapM (lane 1) was loaded on both columns. The proteins retained at resin were eluted with glutathione (lanes 4 and 5). The bottom panel shows the Western blotting confirmation of His-PapM presence in eluate in the presence of GST-ParA but not in control GST eluate. Anti-His antibody was used for His-PapM detection. Presented result is the representative example of independent four experimental repeats.

with the following previously described ParA variants: ATP binding deficient monomer ParA$_{K44A}$; dimerization-impaired ParA$_{G40V}$; ATP hydrolysis deficient and DNA-associated dimer ParA$_{D68A}$; DNA interaction deficient dimer ParA$_{R219E}$; and ParA$_{T3A}$, which is unable to interact with DivIVA (37). The only T25-ParA variant that interacted with T18-PapM was ParA$_{R219E}$, suggesting that PapM preferably binds ParA dimer in the absence of DNA (Fig. S1A). This notion was reinforced by the lack of detected interactions with monomer version ParA$_{K44A}$ and ParA$_{G40V}$ (Fig. S1A). Interestingly, an alphafold multimer model of a dimer of PapM and a dimer of ParA (40) predicted a PapM interaction at the DNA binding

site on ParA, which could indicate PapM interfering with the ParA-DNA interaction (Fig. S1B).

Summarizing, the BTH library screening identified a novel ParA partner named PapM, while an affinity chromatography pulldown approach confirmed a direct interaction between ParA and PapM.

## PapM, ParA, and DivIVA form a tripartite complex, which is affected by DivIVA phosphorylation

In mycobacteria, ParA interacts, apart from ParB, also with DivIVA (34, 37). Therefore, we tested whether PapM could interact with other ParA partners. Indeed, BTH analysis revealed T18-PapM interaction with T25-DivIVA but not with T25-ParB (Fig. 1A). Further testing of the PapM interactions using *M. smegmatis* DivIVA subdomains fused to the T18 domain of adenylate cyclase indicated that interaction interface may encompass the linker between DivIVA coiled-coil domains and fragment of second coiled coil, the region that contains a single phosphorylation site—Thr 74. Interestingly, the same fragment of DivIVA was earlier shown to be involved in the interactions with ParA (34).

Next, based on the obtained BTH results, we set out to examine PapM interactions with ParA and DivIVA using protein colocalization assays in *E. coli*. Earlier, we have shown that EGFP-ParA colocalized at the poles of *E. coli* cells with mCherry-DivIVA when coproduced (9, 37). Here, we tested whether PapM fused to fluorescent protein mTurquoise2 (PapM-mT2) would colocalize with the polar mCherry-DivIVA complex and whether ParA recruitment to polar DivIVA complex would be affected by the presence of PapM-mT2. Additionally, we investigated whether DivIVA phosphorylation would influence ParA or PapM interactions.

In *E. coli* cells, PapM-mT2 localized at the poles in the presence of polarly localized mCherrry-DivIVA but not in the control strain with polarly localized Ics-mCherry [Ics protein fragment targeted the mCherry to the cell poles (41)] (Fig. 2A and B). In *E. coli* coproducing PapM-mT2, EGFP-ParA, and mCherry-DivIVA, the three fusion proteins colocalized at the cell poles (Fig. 2D). This result indicates the engagement of PapM in the formation of the tripartite complex. Measurements of the ratio of EGFP-ParA fluorescence at the poles to its fluorescence at the mid-cell indicated that EGFP-ParA recruitment to mCherry-DivIVA was not visibly affected by the presence of PapM-mT2 (Fig. 2I).

Next, using the same *E. coli* colocalization approach, we set out to examine if DivIVA interaction with ParA and PapM may be affected by DivIVA phosphorylation. We confirmed DivIVA phosphorylation by coproduction with either His-PknA (His-PknA$_{KD}$) or His-PknB kinase domain (His-PknB$_{KD}$) in *E. coli* (Fig. S2). Microscopy analysis showed that, in the presence of His-PknB$_{KD}$, recruitment of EGFP-ParA to mCherry-DivIVA was diminished and an increased EGFP-ParA fluorescence signal along the cell was observed (Fig. 2C, E, and I). Markedly, when mCherry-DivIVA was produced in the presence of both His-PknB$_{KD}$ and PapM-mT2, the signal from EGFP-ParA was fully dispersed along the cell, while PapM-mT2 still colocalized with mCherry-DivIVA (Fig. 2F and I). Next, using phosphoablative mCherry-DivIVAT74A variant, we found that EGFP-ParA fully colocalized with this variant in the presence of His-PknB$_{KD}$ (Fig. 2G). This analyses confirmed that colocalization of mCherry-DivIVA and EGFP-ParA is disrupted by DivIVA phosphorylation.

The analysis of the ratio of EGFP-ParA fluorescence at the poles to its fluorescence at the mid-cell confirmed that PapM reduced the polar EGFP-ParA recruitment to mCherry-DivIVA in the presence of PknB kinase domain (Fig. 2E, F, and I). Thus, PapM enhanced the DivIVA phosphorylation-dependent release of ParA from the polar DivIVA complex. We also noted that, in the presence of His-PknB$_{KD}$ (in the absence of ParA), the ratio of PapM-mT2 fluorescence at the pole to its fluorescence at the mid-cell increased (Fig. 2 B, H, and I). This indicated that PapM-mT2 colocalized more efficiently with phosphorylated mCherry-DivIVA than with unphosphorylated.

Additionally, we tested whether PapM-mT2 colocalized with nucleoid-bound EGFP-ParA in *E. coli* (in the absence of mCherry-DivIVA). The nucleoid staining confirmed that EGFP-ParA colocalized with DNA in the absence and the presence of PapM-mT2

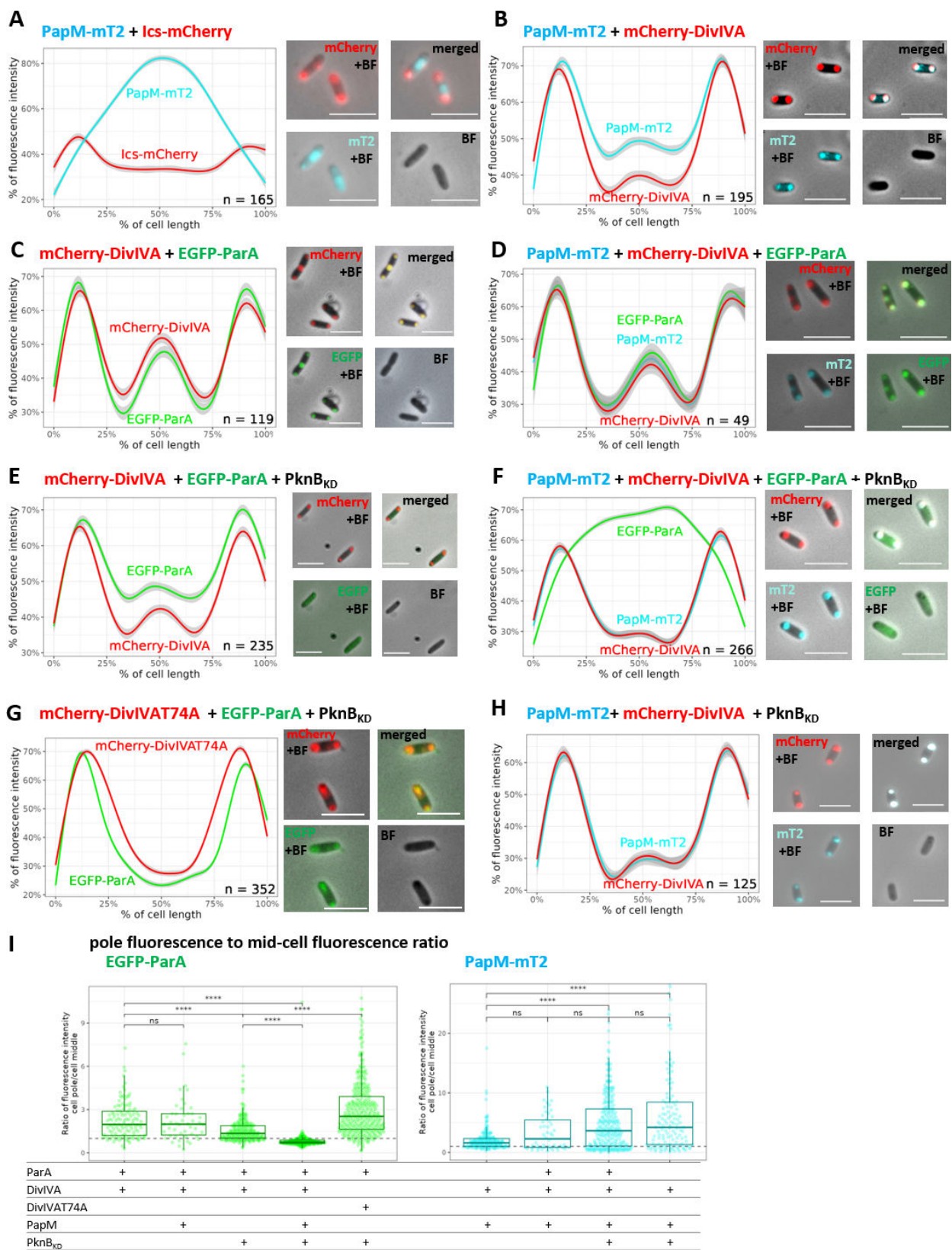

**FIG 2** Phosphorylation enhances PapM binding to DivIVA while diminishing ParA recruitment to DivIVA in *E. coli*. (A) Colocalization of PapM-mTurquoise2 (MT2) with Ics-mCherry. (B) Colocalization of PapM-mT2 with mCherry-DivIVA. (C) Colocalization of mCherry-DivIVA and EGFP-ParA. (D) Colocalization of mCherry-DivIVA, EGFP-ParA and PapM-mT2. (E) Colocalization EGFP-ParA with mCherry-DivIVA in presence of His-PknB_KD. (F) Colocalization of mCherry-DivIVA, (Continued on next page)

**FIG 2** (Continued)

EGFP-ParA, and PapM-mT2 in the presence of His-PknB$_{KD}$. (G) Colocalization of phosphoablative mCherry-DivIVAT74A variant and EGFP-ParA in the presence of His-PknB$_{KD}$. (H) Colocalization of PapM-mT2 with mCherry-DivIVA in the presence of His-PknB$_{KD}$. Left panels show the profiles of fluorescence measured along the *E. coli* BL21 cells, and the right panels show the representative images of *E. coli* cells producing EGFP-ParA (green), mCherry-DivIVA (red), and PapM-mT2 (blue). The number of the cells used for analysis (n) is indicated in the plots; scale bar, 5 μm. (I) The ratio of EGFP-ParA (left panel) and PapM-mT2 (right panel) fluorescence at the pole to the fluorescence at the mid-cell. The ratio higher than 1 (dashed line) means more fluorescence signal at the poles; the ratio lower than 1 illustrates the fluorescence in the mid-cell. The data come from at least two independent biological replicates. The statistical significance between strains determined by Wilcoxon test (two-sided) with Holm method used for multiple comparisons is marked with asterisks: *P*-values ≤0.05 (*), ≤0.01 (**), ≤0.001 (***), and ≤0.0001 (****).

(Fig. S3A through D). PapM-mT2 colocalized with EGFP-ParA and the nucleoid (Fig. S3D). Finally, since PapM was annotated as the protein belonging to TetR regulators family, we also tested PapM interactions with DNA. The electrophoretic mobility shift assay (EMSA) experiment with supercoiled pUC19 plasmid did not indicate PapM-DNA binding (Fig. S4A), which could be due to structural alteration within the putative HTH motif in PapM or the lack of specific binding site in the tested DNA (Fig. S4B and C).

To summarize, the *E. coli* colocalization approach confirmed the engagement of PapM in the interaction between ParA and DivIVA. This analysis also indicated that ParA recruitment to DivIVA was diminished by DivIVA phosphorylation, while PapM recruitment was boosted. We also found that PapM enhanced the phosphorylation-dependent release of ParA from the polar DivIVA complex.

## Modifications of *papM* levels affect the *M. smegmatis* growth, chromosome segregation, and cell length in a ParA-dependent manner

Having established that PapM interacts directly with ParA, we set out to investigate the biological function of PapM in *M. smegmatis*. To this end, we constructed *M. smegmatis* strains with *papM* deletion and a *papM* overexpression in the wild-type and Δ*parA* background. Since *parA* deletion was earlier shown to slow the *M. smegmatis* culture growth, leading to the formation of anucleate cells and altering the cell length (34), we examined the impact of modified *papM* expression on the above-mentioned processes.

Deletion of *papM* in the wild-type background did not affect the growth rate; however, *papM* deletion in the Δ*parA* background slowed the growth of the mutant strain in comparison to Δ*parA* parental strain (Fig. 3A). Next, we used the pMVp$_{ami}$*papM* construct in which *papM* was placed under inducible but leaky p$_{ami}$ promoter [which triggers expression at a significant level even without adding inducer (Fig. S4A)]. This construct introduced to the Δ*parA*Δ*papM* background successfully complemented the *papM* deletion (Fig. S5B). Using the pMVp$_{ami}$*papM* construct, we further tested the growth of the strains with an increased *papM* expression in wild-type and Δ*parA* genetic background. In absence of an inducer, the overexpression of *papM* in the wild-type background [at approximately three to five times level of the wild type (Fig. S5A)] did not alter the growth rate in comparison to the wild-type control strain (with empty pMVp$_{ami}$) (Fig. 3B). Also, upon addition of the inducer to the *papM*, overexpressing strain did not affect the growth rate compared to wild-type control strain (Fig. S5C). Interestingly, the increased level of *papM* expression in the Δ*parA* background fully restored the growth rate of the parental Δ*parA* strain to the growth rate of wild-type control strain (Fig. 3B; Fig. S4C). The overexpression of *papM* in the Δ*parA*Δ*parM* background also restored the even slower growth rate of the parental double deletion strain confirming full complementation of *papM* deletion.

Next, we investigated whether *papM* deletion or overexpression affected chromosome segregation. We found that *papM* deletion in the wild-type background led to a modest chromosome segregation defect (5.6% anucleate cells in Δ*papM* compared to 0.6% in the wild-type strain) (Table 1). Deletion of *papM* in the Δ*parA* background had negligible influence on the segregation defect resulting from the lack of ParA (31.9% anucleate cells in the Δ*parA*Δ*papM* strain, compared to 30.8% of anucleate cells in the Δ*parA* strain); however, changes of the cell shape, such as bulging or branching, were

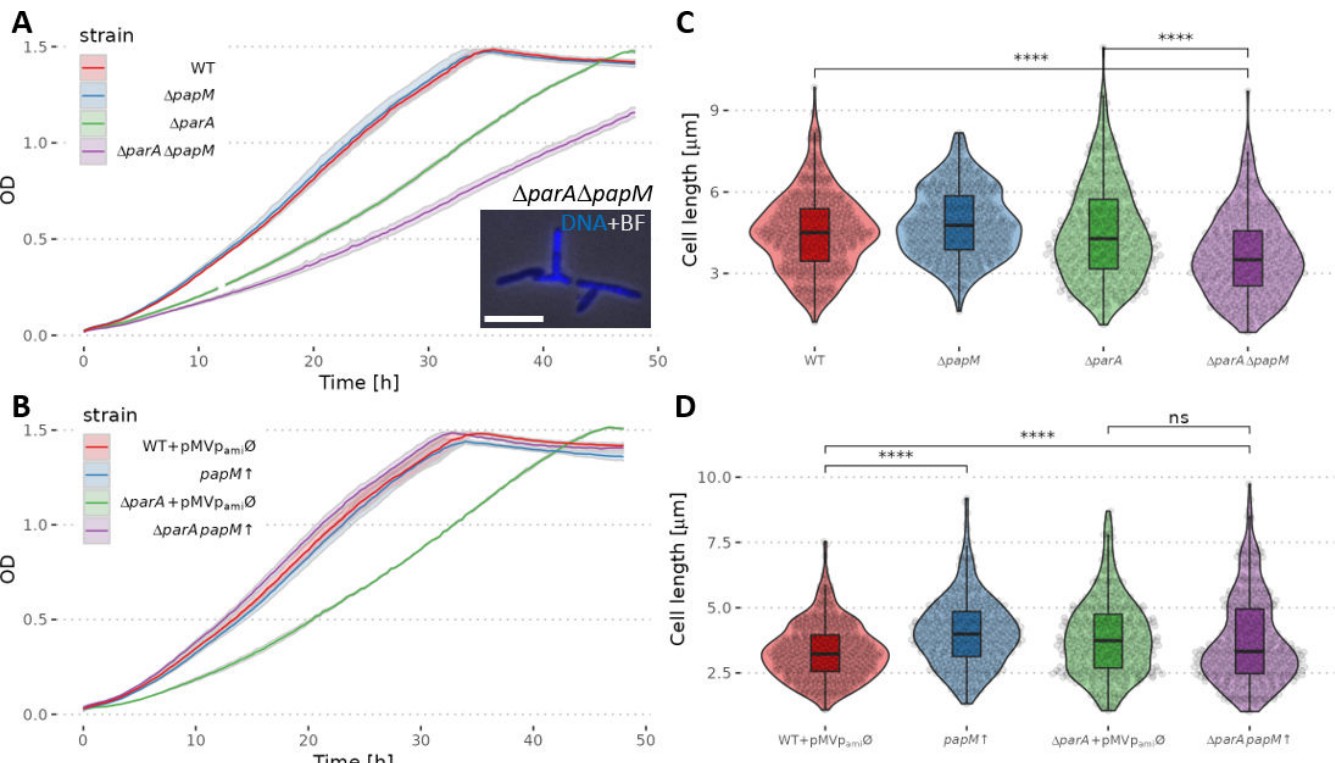

**FIG 3** *papM* deletion or its overexpression in wild-type and Δ*parA* background affects *M. smegmatis* culture growth rate and cell length. (A) The growth curves of Δ*papM*, Δ*parA*, and Δ*parA*Δ*papM* strain compared to the wild-type (WT) strain; inset, example image of branched Δ*papM*, Δ*parA M. smegmatis* cells. The image shows the overlay of DAPI fluorescence (nucleoid stain, blue) and brightfield. Scale bar, 5 µm. (B) The growth curve of *papM* overexpressing strain in wild-type (*papM*↑) and Δ*parA* background (Δ*parA papM*↑) as compared to control strains with empty pMVp_ami plasmid (WT + pMVp_amiØ and Δ*parA* + pMVp_amiØ). (C) Cell length distribution in Δ*papM* (478 cells), Δ*parA* (346 cells), and double mutant Δ*parA*Δ*papM* (350 cells) strain, compared to wild-type strain (WT) (482 cells). (D) Cell length distribution in *papM* overexpressing strain in wild-type (433 cells) or Δ*parA* background (392 cells) as compared to controls with empty pMVp_ami vector in the wild-type (488 cells) or Δ*parA* background (351 cells). The data come from at least two independent biological replicates. The statistical significance between strains determined by Student's *t*-test (two-sided) with Holm method used for multiple comparisons is marked with asterisks: *P*-values ≤0.05 (*), ≤0.01 (**), ≤0.001 (***), and ≤0.0001 (****).

detectable (Fig. 3A, inset). A significant disturbance of chromosome segregation was caused by *papM* overexpression in the wild-type background (13.4% of anucleate cells). Surprisingly, the segregation phenotype resulting from *papM* overexpression seemed to be not reflected in the growth rate of the mutant strain. Notably, the overexpression of *papM* in Δ*parA* partially restored the chromosome segregation defect caused by deletion of *parA*, lowering the fraction of anucleate cells when compared to the parental strain (21.6% at *papM* overexpression in Δ*parA* background compared to 30.8% in parental Δ*parA* strain). The PapM-dependent reduction of chromosome segregation defect in the Δ*parA* background was consistent with the observed increased growth rate of the strain overexpressing *papM* in the Δ*parA* background.

The chromosome segregation defect resulting from *parA* deletion was earlier noted to be reflected by the altered cell length (34). The increased variation of the cell length was attributed to mispositioned cell division and the formation of short, anucleate

**TABLE 1** Chromosome segregation defects and cell length in Δ*papM* strain or *papM* overexpression in wild-type (*papM*↑) or Δ*parA* background (Δ*parA papM*↑)

|  | WT | Δ*papM* | Δ*parA* | Δ*parA* Δ*papM* | WT pMVØ | *papM*↑ | Δ*parA* pMVØ | Δ*parA, papM*↑ |
|---|---|---|---|---|---|---|---|---|
| Anucleate cells % | 0.6 | 5.6 | 30.8 | 31.9 | 2.2 | 13.4 | 29.2 | 21.6 |
| Cell length mean ± 95% CI, median | 4.52 | 4.85 | 4.51 | 3.66 | 3.32 | 4.05 | 3.86 | 3.74 |
|  | ± 0.1, | ± 0.1, | ± 0.14, | ± 0.13, | ± 0.09, | ± 0.11, | ± 0.13, | ± 0.13, |
|  | 4.5 | 4.77 | 4.28 | 3.51 | 3.23 | 3.99 | 3.73 | 3.32 |

cells, while the length of the other daughter cells increased. Markedly, the abolished interaction between ParA and DivIVA also increased the rate of cell elongation (37). Therefore, here, we tested whether *papM* deletion or its overexpression in wild-type and *parA* deletion background influenced the cell length.

While *papM* deletion in the wild-type background did not affect the average cell length, *papM* deletion in the Δ*parA* background resulted in the decreased average cell length (Fig. 3C; Table 1). Elevated levels of PapM increased the cell length in the wild-type background when compared to the wild-type control with an empty pMVp$_{ami}$ vector. However, in the Δ*parA* background, moderate PapM overproduction did not affect cell length significantly (Fig. 3D; Fig. S5D; Table 1).

To summarize, the analysis of the *M. smegmatis* mutant strains showed that PapM affected chromosome segregation and influenced cell length. PapM involvement in the same cell cycle processes as ParA confirms a functional relationship between PapM and ParA. Notably, the absence of PapM also affected the growth and the cell shape of the strain lacking ParA, which may indicate its direct impact on DivIVA.

## Deletion of *papM* alters ParA dynamics

Given that PapM affected ParA-DivIVA interaction in *E. coli*, to further investigate the impact of PapM on ParA, we analyzed whether the deletion of *papM* influences ParA localization in *M. smegmatis*.

Earlier studies showed dynamic EGFP-ParA localization patterns during the *M. smegmatis* cell cycle (35). In the new-born cells, EGFP-ParA fluorescence signal was first visible at the new cell pole; next, it was dynamically extended as a cloud-like fluorescence along the cell. At the cell division time, ParA was relocated to the mid-cell, being established at the newly formed pole (35). Here, we performed time-lapse fluorescent microscopy analysis of EGFP-ParA during the cell cycle in a *papM* deletion background (Δ*parA*Δ*papM* complemented with the pMV306p$_{nat}$*egfp-parA*) and in the control strain (Δ*parA* complemented with the pMV306p$_{nat}$*egfp-parA*). This was followed by measurements of EGFP-ParA fluorescence intensity along the cell during the cell cycle (from the time of cell separation to the progeny cell separation).

The time-lapse analysis showed that the ParA relocation pattern was affected by deletion of *papM*. The kymograph analysis of average fluorescence intensity along the cell indicated that, in the control strain, EGFP-ParA relocation to the mid-cell position was detected in about 60%–70% of the cell cycle (Fig. 4A, red ellipse, Fig. S6; Movie 1A). In the Δ*papM* strain, accumulation of EGFP-ParA signal at the mid-cell preceding cell division was less clearly detectable (Fig. 4A; Fig. S6; Movie 1B). This may indicate that either the EGFP-ParA signal was more dispersed along the cell or its relocation was less tightly coordinated during the cell cycle.

Next, to further determine the impact of PapM on ParA dynamics, we used photoactivation localization microscopy (PALM) to track PAmCherry-ParA molecules (in Δ*parA* strain complemented with the pMV306p$_{nat}$*PAmcherry-parA*) and to determine if their mobility was affected by *papM* deletion. Earlier, it was shown that mutations that affect ParA binding to DNA (but not to DivIVA) had a significant impact on ParA mobility (37).

Here, we found that the elimination of PapM increased the fraction of immobile PAmCherry-ParA molecules (from 57.4% to 70.9% of molecules with diffusion coefficient $d = 0.06~\mu m^2~s^{-1}$) and decreased the fraction of mobile molecules (from 42.6% to 29.1% of molecules with diffusion coefficient $d = 0.48~\mu m^2~s^{-1}$), decreasing the average diffusion coefficient from $0.25~\mu m^2~s^{-1}$ in wild-type background to $0.20~\mu m^2~s^{-1}$ in the absence of PapM (Fig. 4B).

To sum up, these two experiments suggested an impact of PapM on ParA dynamics. Deletion of *papM* altered the cell cycle-dependent ParA relocation to the newly established cell poles and lowered the mobility of ParA molecules. This altered mobility could possibly result from modified ParA association with DNA which was earlier shown to predominantly affect ParA mobility (37).

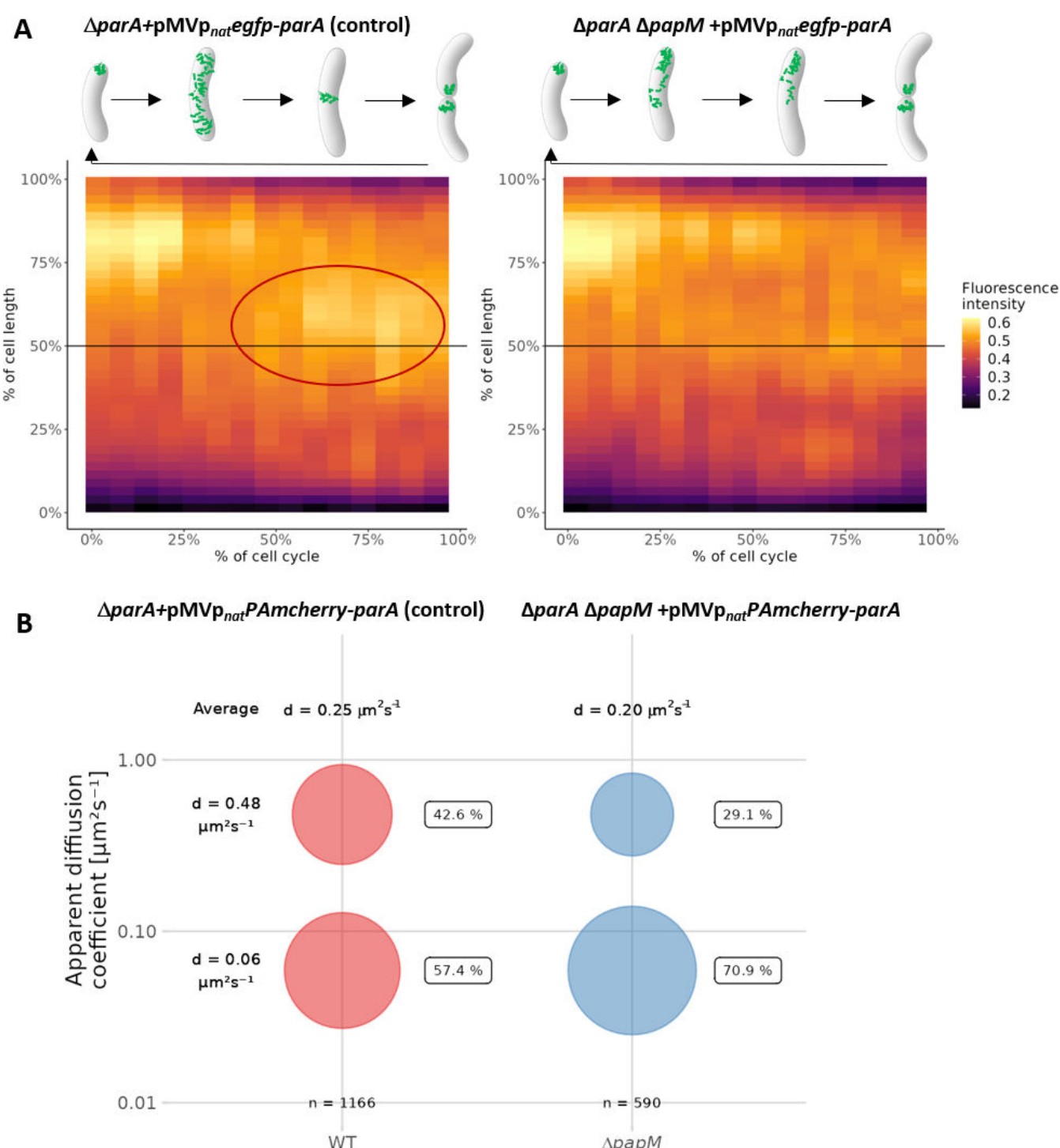

**FIG 4** *papM* deletion alters ParA dynamics. (A) Kymographs showing cumulative EGFP-ParA fluorescence intensity during the cell cycle in the control strain (Δ*parA* + pMVp$_{nat}$*egfp-parA*, 30 cells analyzed) and in the Δ*papM* background (Δ*parA*Δ*papM* + pMV306p$_{nat}$*egfp-parA*, 25 cells analyzed). The time zero is the beginning of the new cell cycle detected as the visible separation of daughter cells accompanied by EGFP fluorescence at the new poles. Mid-cell is marked with a black line, and the ParA assembly at the mid-cell in the control strain is marked with red ellipse. The data come from two independent biological replicates. (B) The bubble plot showing the percentages of mobile and immobile PAmCherry-ParA molecules in control (Δ*parA* + pMVp$_{nat}$*PAmcherry*-parA) and Δ*papM* (Δ*parA*Δ*papM* + pMV306p$_{nat}$*PAmchery-parA*) strain as determined by PALM. The number of tracks analyzed is 1,166 for control (49 cells) and 590 tracks (42 cells) for Δ*papM*. The diffusion coefficients are indicated.

## Deletion of *parA* or *papM* modifies the subcellular distribution of DivIVA, affecting the cell elongation and division

Earlier, we demonstrated that impairing ParA-DivIVA interaction altered the cell elongation rate, which indicated the influence of ParA on DivIVA activity in *M. smegmatis* (37). Our *E. coli* colocalization studies presented above indicated that PapM interacted with DivIVA and influenced ParA-DivIVA interaction in the presence of the kinase domain. Additionally, we observed that changes in PapM levels altered the *M. smegmatis* cell length, which could be due to PapM direct or indirect influence on DivIVA. To investigate PapM's impact on DivIVA and to compare it to the effect of ParA, we analyzed the polar and mid-cell accumulation of mCherry-DivIVA in Δ*parA* and Δ*papM* strains. Moreover, using the time-lapse microscopy, we analyzed the cell elongation rate in a *parA* and *papM* deletion background.

Analysis of mCherry-DivIVA accumulation at the cell poles showed that deletion of *parA* increased the intensity of the mCherry-DivIVA polar fluorescence signal as compared to fluorescence intensity at the mid-cell (Fig. 5A, B, and D). The increase of mCherry-DivIVA fluorescence was more pronounced at the old pole in comparison to the new pole (Fig. 5A, B, and D). The effect of *papM* deletion was less significant than that of *parA* deletion (Fig. 5A, B, and D). Moreover, while *papM* deletion had no detectable effect on mCherry-DivIVA fluorescence asymmetry, *parA* deletion increased the ratio of mCherry-DivIVA fluorescence intensity at the old pole to its fluorescence intensity ad the new pole, therefore enhancing the asymmetry of *M. smegmatis* cells (Fig. 5C).

The differences in DivIVA signal accumulation could be related to the modified cell elongation rate. Therefore, we used time-lapse microscopy to analyze the cell cycle parameters. We measured the cell cycle length of the strains producing mCherry-DivIVA (apart from the wild-type protein) as the time between the appearance of the mCherry-DivIVA signal in the center of the mother cell and the appearance of the mCherry-DivIVA fluorescence in the middle of the daughter cell. We further measured the cell elongation rate of the strains producing mCherry-DivIVA. Additionally, visualization of peptidoglycan with NADA allowed us to detect cell division and measure the cell elongation rate.

The analysis of the cell cycle parameters clearly indicated that *parA* deletion increased the cell elongation rate and the acceleration was more significant for the old pole inheriting cell (Fig. 6A; Fig. S7). The cell elongation rate of a Δ*parA* strain was also more varied in comparison to the wild-type control strain, but the average length of the cell cycle was not affected (Fig. 6A and B; Fig. S7). The increased cell elongation rate was consistent with the occurrence of the elongated cells that were observed in the Δ*parA* strain. Elimination of *papM* somewhat increased variation of the cell extension rates but did not visibly modify the average length of the cell cycle (Fig. 6A and B). Analysis of Δ*parA*, Δ*papM*, and wild type stained with fluorescent D-amino acid (NBD-amino-d-alanine, NADA) indicated a higher cell extension rate than extension rate of the respective strains producing mCherry-DivIVA (Fig. S7) which may be due to disturbances caused by the production of fusion mCherry-DivIVA apart from wild-type DivIVA. However, the influence of *parA* deletion and *papM* deletion on growth rate was the same as in the strains producing mCherry-DiviVA (Fig. S7).

Next, we focused on the mid-cell mCherry-DivIVA signal which indicated the cell division and DivIVA relocation to the newly formed septum. Time-lapse microscopy allowed us to measure the timespan between the mid-cell mCherry-DivIVA appearance and the visible separation of the two fluorescence foci indicating the establishment of the two daughter cell poles. In the wild-type strain, the mCherry-DivIVA signal was visible for a shorter time than in the Δ*parA* and Δ*papM* strains (69 min on average in wild-type control strain, in comparison to 80 min in the Δ*parA* strain and 75 min in Δ*papM* strain) (Fig. 6C; Movie 2A through C). This analysis was reinforced by timepoint analysis of the mCherry-DivIVA producing strains: in both *parA* and *parM* deletion strains, the fraction of cells with the mid-cell mCherry-DivIVA fluorescence signal increased (from 14% in wild type to 24% in Δ*parA* and 19% in Δ*papM*, respectively) (Fig. 6D). This confirms that

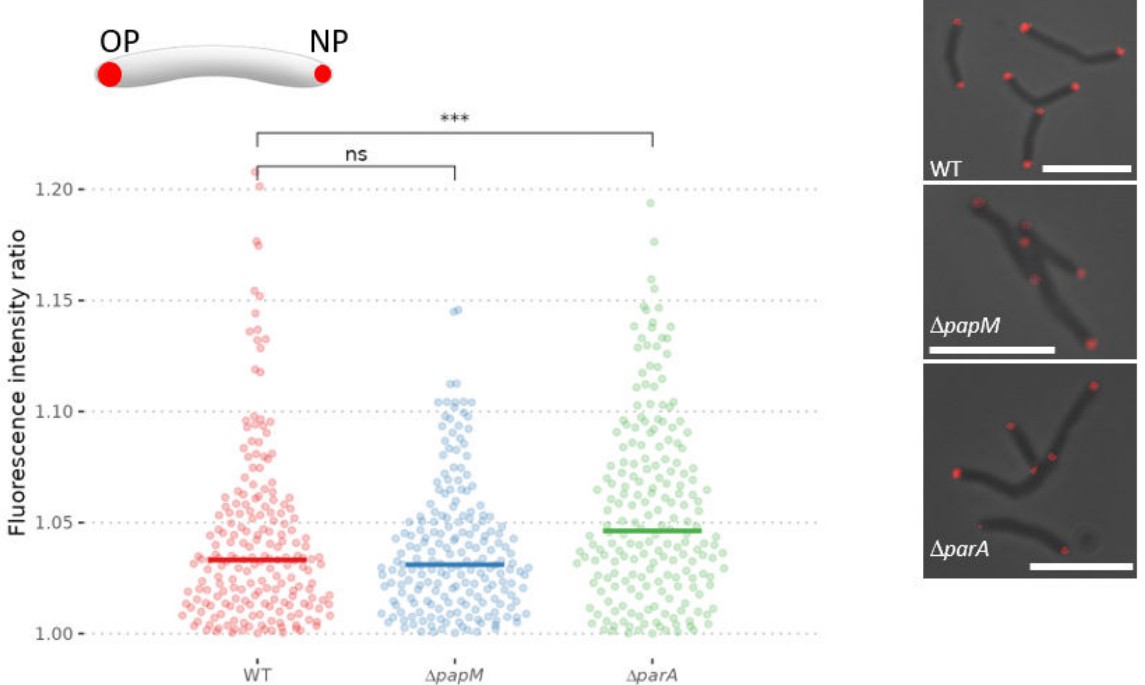

**FIG 5** Polar accumulation of mCherry-DivIVA is altered in ΔparA but not ΔpapM M. smegmatis strain as compared to the wild-type control. The analyses were performed in WT control strain (400 cells), ΔpapM (398 cells) and ΔparA (400 cells) strains producing mCherry-DivIVA (apart of the wild-type DivIVA). (A) The ratio of mCherry-DivIVA fluorescence intensity at the old pole to average fluorescence intensity in the mid-cell. (B) The ratio of fluorescence intensity at the new pole

**FIG 5** (Continued)

to average fluorescence intensity in the mid-cell. (C) The ratio of mCherry-DivIVA fluorescence intensity at the old cell pole to its fluorescence at the new cell pole. (D) Representative images of the cells of each studied *M. smegmatis* strain. The mCherry-DivIVA fluorescence (red) is merged with the brightfield image. Scale bar, 5 µm. The data come from at least two independent biological replicates. The statistical significance between strains determined by Wilcoxon test (two-sided) with Holm method used for multiple comparisons is marked with asterisks: *P*-values ≤0.05 (*), ≤0.01 (**), and ≤0.001 (***).

the timespan required to establish the new pole is increased in both Δ*parA* and Δ*papM* strains.

To summarize, analysis of mCherry-DivIVA fluorescence implied that the absence of ParA increased the DivIVA accumulation at the old pole and increased cell extension rate. Elimination of ParA also affected DivIVA relocation to the cell division site and establishment of the new poles. Deletion of PapM had a marginal impact on DivIVA accumulation at the poles but influenced DivIVA redistribution to the mid-cell. This reinforces the PapM impacts on DivIVA localization and is consistent with the fact that modified PapM levels exhibit an effect on cell length.

## DISCUSSION

We have identified a novel ParA and DivIVA partner—named PapM—that contributes to the cell cycle regulation in *M. smegmatis*. Notably, while in other bacteria, ParA was shown to be engaged in interactions with polar or subpolar proteins, i.e., TipN and PopZ in *C. crescentus*, HubP in *V. cholerae* or Scy in *S. coelicolor* (4, 42–45), according to our best knowledge, our findings are the first report of the third partner that interferes with these interactions.

The PapM homologues were found only in several *Mycobacterium* species, mostly those closely related to *M. smegmatis*, while the protein has no close homologues in either *M. tuberculosis* or in many other mycobacteria. This suggest that *papM* gene could have been eliminated in pathogenic, intracellular mycobacteria whose genomes undergo reduction and which are not exposed to dramatic changes of environmental conditions. That also raises the question if PapM role could be related to fast growth and adjustment to changes of environmental conditions. However, it should be noted that the elimination of PapM had no detrimental effect on the *M. smegmatis* growth rate under optimal conditions. Interestingly, although PapM was annotated as a protein from the TetR regulator family, it did not show DNA interaction in an EMSA assay. That could be explained either by high specificity toward unknown DNA sequence or by weak DNA binding. Intriguingly, PapM HTH motif analysis and structure prediction indicated structural changes within HTH motif which could interfere with DNA binding. This could also indicate that PapM evolved to play different role than transcriptional regulator.

We show that PapM modulates the interplay between ParA-DivIVA. Our *E. coli* colocalization analysis indicated that PapM promotes the release of ParA from DivIVA in presence of the PknB kinase domain, which phosphorylates DivIVA. The impact of PapM on ParA dynamics was confirmed by EGFP-ParA localization and the single molecule tracking in *M. smegmatis* cells. The decreased mobility of ParA molecules in absence of PapM may be due to enhanced binding of ParA to the nucleoid or the DivIVA complex. Earlier, it was shown that the ParA$_{D68A}$ variant that is not able to hydrolyze ATP had significantly reduced mobility due to enhanced nucleoid binding, while the variant ParA$_{R219E}$ had increased mobility due to abolished interaction with nucleoid (37). The detailed analysis of PapM interaction with ParA variants suggested the preferred interaction with ParA dimer that is not able to bind DNA (ParA$_{R219E}$). The impact of PapM on ParA interactions with DNA and protein dynamics may account for the observed segregation phenotype of PapM overproducing strain. Elevated levels of PapM could interfere with ParA-DNA and/or ParA-DivIVA interaction affecting ParA dynamics and impairing segrosomes separation.

Markedly, PapM also interacts with DivIVA. *E. coli* colocalization analysis suggested that PapM recruitment to DivIVA is enhanced in presence of ParA. Interestingly, the analyses of interactions using DivIVA subdomains suggested that the DivIVA fragment

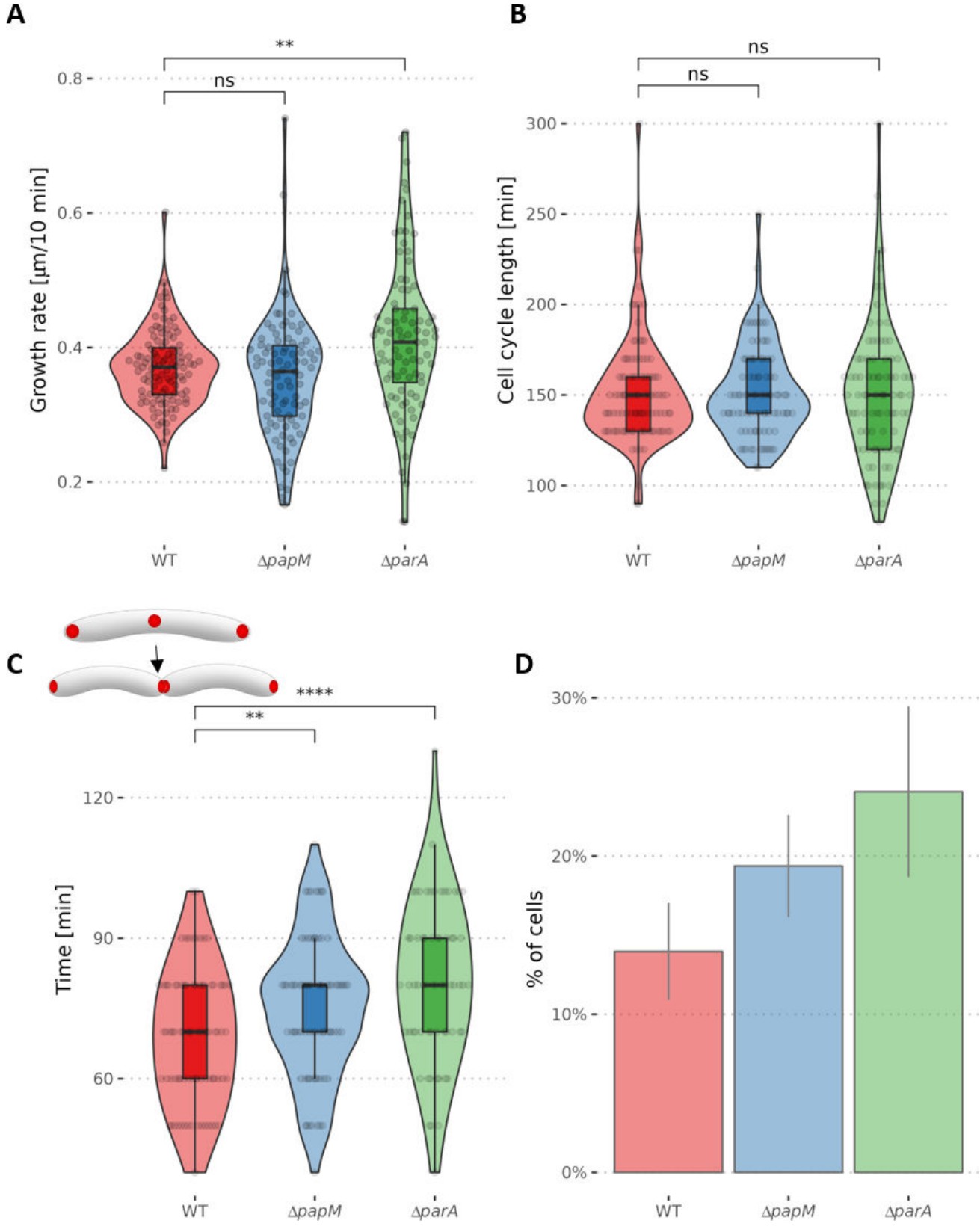

**FIG 6** The growth rate and mCherry-DivIVA redistribution during the cell cycle are affected by *parA* and *papM* deletion. (A) The cell growth rate measured as the cell length increment (the difference of the cell length of the newly born mother cell and the daughter cell at the time preceding cell division - the mid-cell mCherry-DivIVA signal appearance) divided by cell cycle length shown in B (103, 90, and 89 cells analyzed for WT, Δ*papM*, and Δ*parA*, respectively). (B) The cell

**FIG 6** (Continued)

cycle length - the time elapsed between the appearance of the mid-cell mCherry-DivIVA signal in the mother cell and the appearance of mid-cell mCherry-DivIVA signal in its daughter cell (the number of cells analyzed as in A). (C) The time between the detection of the mid-cell mCherry-DivIVA signal and detection of the separated cell poles (84, 78, and 65 cells analyzed for WT ΔpapM and ΔparA, respectively). The analyses were performed in the WT control strain and ΔpapM and ΔparA strains producing mCherry-DivIVA (apart of the wild-type DivIVA). (D) The percentage of the cells with the visible mCherry-DivIVA signal in the snapshot images (487, 573, and 241 cells analyzed for WT, ΔpapM, and ΔparA, respectively). The data come from two independent biological replicates. The statistical significance between strains determined by Student's t-test (two-sided) with Holm method used for multiple comparisons is marked with asterisks: P-values ≤0.05 (*), ≤0.01 (**), ≤0.001 (***), and ≤0.0001 (****).

that encompasses phosphorylation site Thr74 potentially may be involved in PapM binding. This implies the influence of DivIVA phosphorylation on the interactions with PapM. Indeed, PapM colocalization with DivIVA was enhanced in the presence of the PknB kinase domain. The impact of PapM on DivIVA was also demonstrated by changes in the cell length in strains with modified PapM levels. Overexpression of *papM* in a wild-type background increased the cell length, while *papM* deletion in a *parA* deletion background shortened average cell length. Markedly, the elimination of both ParA and PapM had deteriorating effect on growth and led to appearance of misshaped, branched cells, suggesting DivIVA mislocalization. Intriguingly, increased PapM levels in a *parA* deletion background also partially suppressed segregation phenotype resulting from the absence of ParA. The direct influence of PapM on DivIVA and its function in coordination of the cell elongation with the cell division are likely to account for the observed phenotypes. The other possible explanation is the impact of PapM on other proteins that contribute to chromosome segregation or cell cycle coordination.

The important finding of this study is the demonstration that ParA-DivIVA interaction is phosphorylation dependent. Taking into account that DivIVA phosphorylation is correlated with fast cell growth (20, 25, 32), the enhanced binding of ParA to non-phosphorylated DivIVA shown here is consistent with our earlier observations. We noted earlier that polar recruitment of ParA may be induced by unfavorable environmental conditions which inhibit cell growth (37). Moreover, we have shown that the absence of ParA increases the polar accumulation of DivIVA and increases the time required to establish the new poles during cell division. This could be resulting from a slower DivIVA accumulation and/or maturation and, speculatively, phosphorylation at the septum. The lack of ParA also enhances the asymmetry of mycobacterial cell poles leading to greater DivIVA accumulation (speculatively increased phosphorylation) at the old cell pole. These observations are consistent with the observed increased cell elongation rate and also the increased variation of the cell extension rate in the absence of ParA. The strain lacking PapM also exhibited increased variation of the cell extension rate. Such a variation is likely to result from increased heterogeneity, i.e., the difference of the growth rate between the old pole and new pole inheriting cell. Since phosphorylated DivIVA is reportedly more active in cell wall synthesis (25), we are inclined to suggest that ParA, similarly to PapM, may affect DivIVA phosphorylation or, alternatively, interaction with its other protein partners.

Based on the presented results, we propose the speculative model of tripartite interactions (Fig. 7). According to our model, ParA preferably binds DivIVA at the nascent septum. The maturation of the DivIVA complex and its phosphorylation release ParA from the new pole to promote chromosome segregation. On the other hand, ParA has an impact on DivIVA complex stability: in the absence of ParA, "old pole" DivIVA complexes are more stable, and formation of the new pole complexes requires more time. PapM enhances phosphorylation-dependent ParA release from polar DivIVA complex. PapM also has a direct effect on DivIVA maturation but to a lesser extent than ParA. In absence of PapM, formation of DivIVA complex is affected. Altered PapM levels in absence of ParA affect DivIVA function and the cell cycle, possibly interfering with switch between cell elongation and cell division. The impact on this cell cycle checkpoint is likely to explain the PapM interference with ParA-dependent chromosome segregation. Thus, PapM influences both partners activity, affecting ParA dynamics and DivIVA function. The

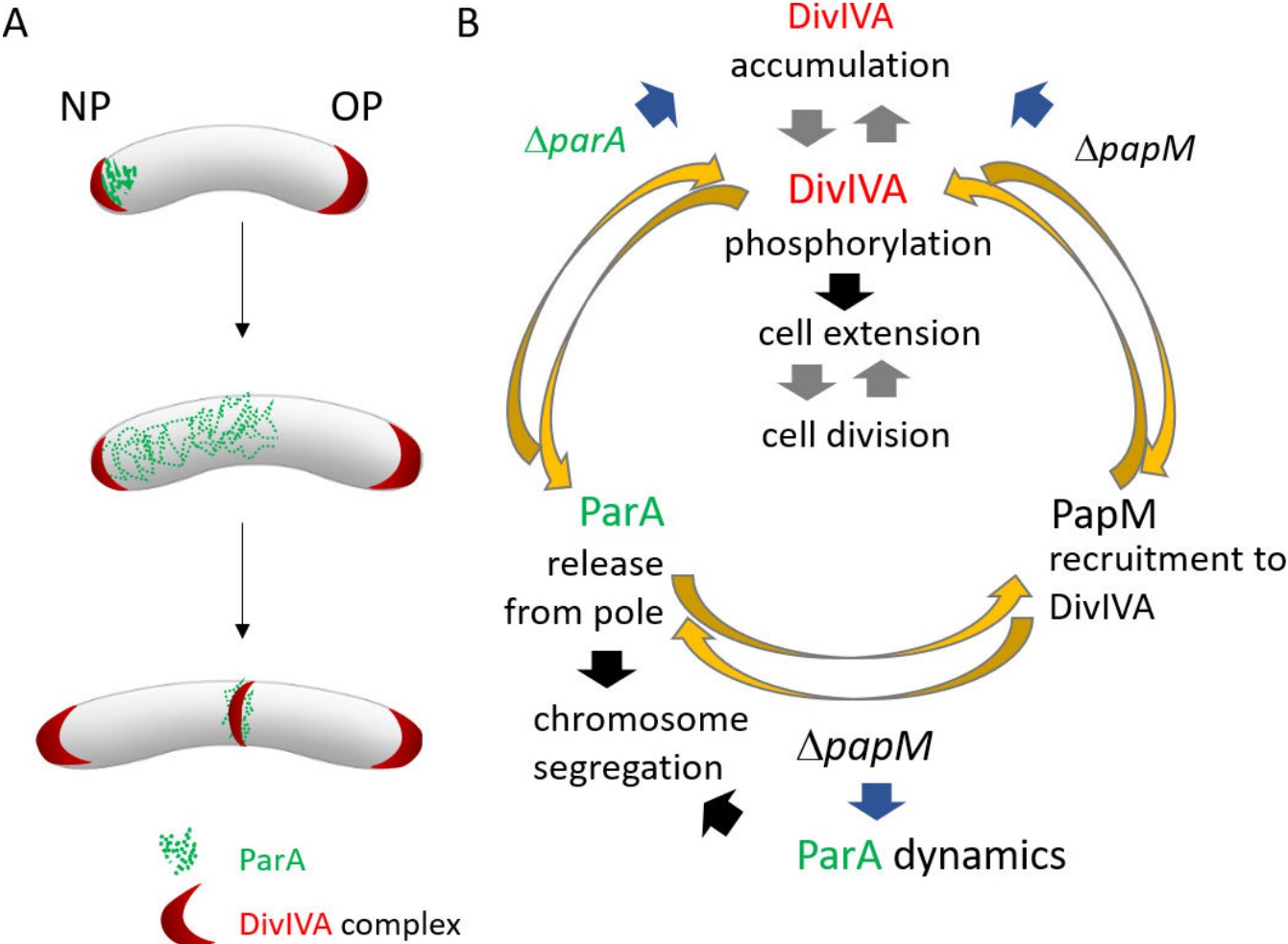

**FIG 7** The model of ParA-PapM-DivIVA interplay. (A) The scheme of *M. smegmatis* cell cycle showing ParA and DivIVA dynamics. (B) Model of ParM-ParA-DivIVA interactions. Black arrows are established relations, gray arrows indicate postulated relations, blue arrows indicate the relations suggested by the mutant *M. smegmatis* strain phenotypic analyses presented here, and yellow arrows illustrate conclusions based on in *E. coli* colocalization experiment.

impact of PapM on DivIVA is likely to be both direct and indirect, by modification of ParA-DivIVA interaction. The tripartite complex interplay seems to be tightly interlinked and coordinates the switch between cell extension and cell division.

## MATERIALS AND METHODS

### Cloning and construct preparation

DNA manipulations were performed using standard protocols (46). Reagents and enzymes were supplied by New England Biolabs (NEB), Merck, and Thermo Scientific. Oligonucleotides were synthesized by Merck and Genomed, and sequencing was performed by Microsynth. *Escherichia coli* strains were grown in lysogeny broth medium at 37°C [DH5α, BL21(DE3)] or 30°C (BTH101). Culture conditions, antibiotic concentrations, and transformation protocols followed standard procedures (46).

### Bacterial two-hybrid system

BTH interaction studies were performed as previously described (38). Details of the construction of the *M. smegmatis* library in the BTH system and the screening of the

library as well as the construction of BTH vectors are described in the Supplementary Material. The plasmids used in this study are listed in Table S1.

## Affinity chromatography

Affinity chromatography was used to test the interaction between PapM and ParA. The GST-ParA cell lysate [40 mL in phosphate-buffered saline (PBS)] was obtained from an 800-mL culture of BL21(DE3) pGEX-6P-2-$parA_{Ms}$ . The clarified lysate was incubated (overnight, at 4°C, with gentle agitation) with 1-mL bed volume of GSH-Sepharose (GE Healthcare). Next, the resin was loaded on the chromatography column, and unbound proteins were removed by washing with cold PBS buffer (at least three resin volumes). In parallel, a control experiment was performed in which the clarified control lysate of BL21(DE3) pGEX-6P-2Φ [prepared according to the same procedure as for BL21(DE3) pGEX-6P-2-$parA_{Ms}$] was loaded on the column. Next, the His-PapM lysate obtained from the 800-mL culture of BL21(DE3) pET28a$papM$ (40 mL of lysate in PBS) was added to the both columns and incubated overnight at 4°C, with gentle agitation. The unbound proteins were removed by washing with at least three-bed volumes of PBS buffer, and then, proteins bound to resin were eluted with PBS buffer containing 20 mM reduced glutathione. All collected fractions (3 mL) were analyzed using SDS-PAGE and Western blotting with an anti-His-tag antibody (Thermo Fisher Scientific).

## *E. coli* colocalization studies

For assays of EGFP-ParA, mCherry-DivIVA, and PapM-mT2 colocalization in *E. coli* BL21(DE3) cells, pACYCDuet-1 vector derivatives containing *papM-mT2* and/or *egfp-parA* genes and pETDuet-1 vector derivatives containing *mcherry-divIVA* or *ics-mcherry* (as the control) and optionally *his-pknB* fragment encoding kinase domain (11–274 aa) were used. *E. coli* BL21(DE3) strains containing either of the above-mentioned constructs were cultured to the log phase in the presence of ampicillin and chloramphenicol. The expression of the cloned genes was induced by the addition of 0.1 mM IPTG for 1 h. After induction, 10 µL of culture was mounted on the microscopic slides, covered with 5 µL of phosphate-buffered saline (PBS)-glycerol (1:1) solution. Cells were examined by a Leica DM6 B fluorescence microscope equipped with a 100× objective with a DFC7000 GT camera. Images were analyzed using LAS X 3.6.0.20104, Fiji, and "R" software (R Foundation for Statistical Computing, Vienna, Austria; https://www.r-project.org/). *E. coli* cells in phase contrast images were automatically detected using a custom ImageJ script, and fluorescence signal was collected along the long axis of each cell. R software was used to normalize and visualize the fluorescence profiles. The fluorescence intensity was measured at the cell poles (5%–20% and 80%–95% of the cell length) and in the middle of the cell (35%%–65% of the cell length for PapM; 25%–35 and 65%%–75% for ParA).

For analysis of PapM-mT2, EGFP-ParA, and nucleoid colocalization, cultures of BL21(DE3) containing pACYCDuet-1 vector derivatives containing *papM-mT2* and/or *egfp-parA* genes were incubated to the log phase in the presence of chloramphenicol. Then, the expression of *papM-mT2* and/or *egfp-parA* genes was induced by the addition of IPTG to a final concentration of 0.1 mM and further culturing for 1 h. At the same time, to inhibit cell division, nalidixic acid was added up to a final concentration of 25 µg /mL (Chai et al., 2014) (47). Next, the cultures were incubated with DAPI (2 µg/mL) for 20 min, centrifuged (5,000 rpm/5 min) and resuspended in PBS buffer. For SYTO 62 staining, 200 µL of the culture was incubated for 15 min with SYTO 62 (0.25 µM) and then mounted on the microscopic slides, dried, and covered with 5 µL of PBS-glycerol (1:1) solution and coverglass. Specimens were examined by a Leica DM6 B fluorescence microscope equipped with a 100× objective. Pictures were analyzed using a custom ImageJ script as described above and the R software, including the ggplot2 package (48).

## *M. smegmatis* growth conditions

*M. smegmatis* strains were cultured either in Middlebrook liquid 7H9 medium (Difco) supplemented with 10% albumin-dextrose-catalase (ADC; BD) and 0.05% Tween 80 or on

solid 7H10 medium supplemented with 10% oleic acid-albumin-dextrose-catalase (BD), 0.5% glycerol, and 0.05% Tween 80.

To determine the growth rate of *M. smegmatis* strain, cultures were inoculated from glycerol stocks and grown to the log phase (OD$_{600}$, 0.3–0.5). Next, the cultures were diluted in fresh medium with or without the addition of acetamide to concentration 0.1% to an OD$_{600}$ of 0.05, and the final volume of 300 µL of diluted culture was loaded into the wells of a Bioscreen C-compatible honeycomb plate. The microplate cultures were incubated at 37°C with continuous shaking using Bioscreen C [Automated Growth Curves Analysis System, Growth Curves (Alab)]. The optical density measurements were taken at 20-min intervals for 2–3 days. The results were analyzed in Excel and visualized using R.

For microscopic experiments, *M. smegmatis* strains from glycerol stocks were used to inoculate starting cultures, which were then grown for 24 h and used for the inoculation of precultures. Next, after being incubated to the log phase (OD$_{600}$ ~0.5), the precultures were diluted in fresh medium to set up cultures with starting OD$_{600}$ ~0.05 and cultured to the desired OD.

## *M. smegmatis* modifications

The *M. smegmatis* strains used in this study are listed in Table S2. The details of the *M. smegmatis* mutant strains construction are described in the Supplementary Materials. All vectors for *M. smegmatis* transformations were prepared in *E. coli* DH5α. To delete *papM* gene in the *M. smegmatis* mc$^2$ 155 chromosome, targeted gene replacement was performed according to the published protocol (49). The construct based on vectors p2Nil (Kan$^R$) and pGOAL17 (*sacB, lacZ*) (49) was prepared according to the strategy described in Supplementary Materials and integrated into the *M. smegmatis* chromosome by homologous recombination. For the construction of *M. smegmatis* complementation and overexpressing strains, derivatives of the mycobacteriophage L5-based integration-proficient vector pMV306p$_{ami}$ were used as described in Supplementary Materials. Quantitative reverse transcription PCR (RT-qPCR) was performed to confirm the overexpression of the *papM* gene in wild-type [MP24 (37) and IM12] and Δ*parA* genetic background (IM11 and IM13) *M. smegmatis* strains (as described in detailed in Supplementary Materials).

## *M. smegmatis* microscopy

For snapshot microscopy, *M. smegmatis* strains were grown to the log phase (OD$_{600}$ 0.4–0.5) in 7H9 medium (as described above). For analysis of the chromosome segregation, cells were incubated with DAPI (2 µg mL$^{-1}$) for 2 h, centrifuged (5,000 rpm/5 min), and resuspended in PBS buffer. For membrane staining, cells were incubated with FM5-95 (0.5 µg/mL; Thermo Fisher Scientific) for 15 min before the nucleoid staining was completed. Next, bacteria were mounted on the microscopic slides, dried, and covered with 5 µL of PBS-glycerol (1:1) solution and coverglass. Snapshots were taken immediately after sample preparation using a Leica DM6 B microscope with a 100× oil immersion objective.

Time-lapse microscopy analyses were performed in a liquid medium using a CellASIC ONIX platform and compatible B04A plates (Merck), as described previously (50, 51). Early log-phase *M. smegmatis* cultures (OD$_{600}$, 0.2–0.4) were grown in liquid 7H9 medium supplemented with 10% ADC and 0.05% Tween 80. All microfluidic experiments were performed under constant pressure (1.5 psi). For NADA-green staining (Torcis Bio-Techne) (52), cells were loaded into the observation chamber and cultured in 7H9 supplemented with ADC and Tween 80 for 1 h. Then, cells were subjected every hour to a 2-min pulse flow of 7H9-ADC-Tween 80 medium supplemented with NADA-green (0.5 mM). Pictures were recorded using differential interference contrast (DIC) and fluorescence channels (490/20 nm excitation filter and 528/38 nm emission filter for GFP and 575/25 nm excitation filter and 632/60 nm emission filter for mCherry) with a CoolSnap HQ2 camera at 10-min intervals using a Delta Vision Elite inverted microscope

equipped with a 100× oil immersion objective and an environmental chamber set to 37°C. All images were analyzed using the Fiji and R software packages (R Foundation for Statistical Computing, Austria; http://www.r-project.org), including the ggplot2 package (48). For mCherry-DivIVA polar signal analysis, fluorescence data were normalized by subtracting the minimal value of fluorescence. Then, the 20% lowest values of fluorescence signal from each cell were averaged to obtain background fluorescence value of each cell. Next, maximum fluorescence signal from each cell pole was divided by the fluorescence background to obtain the ratio of pole/background fluorescence. For ParA-EGFP time-lapse analysis, cells were manually detected in the images (25–30 cells), and a fluorescence signal was collected along the long cell axis. R software was then used to normalize fluorescence intensity and draw a cumulative kymograph of the ParA-EGFP fluorescence signal.

Single-molecule tracking PALM was performed using Zeiss Elyra 7 microscope. Photoactivatable mCherry (PAmCherry) was activated with a 405-nm laser, with excitation at 561 nm. For each experiment, 10,000 frames were recorded with images being taken every 30 ms. Molecule tracking and localization analysis were performed using the Trackmate plugin from the Fiji program and SMTracker based upon MATLAB software (MathWorks, Natic, MA, USA). Cell segmentation was performed using Fiji and Oufti software. Mobile and immobile protein molecules were distinguished by calculating the apparent diffusion coefficient using the Gaussian mixture mode in SMTracker. Only tracks with at least four frames were considered for the analysis. Due to cell confinement and motion blurring, D* is an apparent diffusion coefficient (53).

## ACKNOWLEDGMENTS

We are grateful to Katarzyna Ginda-Mäkelä for the help with *M. smegmatis* library preparation and screening and Jolanta Zakrzewska-Czerwińska for comments on the manuscript.

This work was funded by OPUS grant 2017/27/B/NZ1/00823.

## AUTHOR AFFILIATIONS

[1]Department of Molecular Microbiology, Faculty of Biotechnology, University of Wroclaw, Wroclaw, Poland
[2]Technical University of Wroclaw, Wroclaw, Poland

## AUTHOR ORCIDs

Izabela Matusiak ⓘ http://orcid.org/0000-0002-8617-7565
Agnieszka Strzałka ⓘ http://orcid.org/0000-0002-7092-0609
Martyna Gongerowska-Jac ⓘ https://orcid.org/0000-0002-3395-6610
Monika Pióro ⓘ http://orcid.org/0000-0002-2854-5075
Dagmara Jakimowicz ⓘ http://orcid.org/0000-0002-7857-064X

## FUNDING

| Funder | Grant(s) | Author(s) |
|---|---|---|
| National Science Center, Poland | 2017/27/B/NZ1/00823 | Dagmara Jakimowicz |

## AUTHOR CONTRIBUTIONS

Izabela Matusiak, Investigation, Methodology, Visualization, Writing – review and editing | Agnieszka Strzałka, Data curation, Formal analysis, Methodology, Visualization, Writing – review and editing | Patrycja Wadach, Investigation, Methodology | Martyna Gongerowska-Jac, Investigation | Ewa Szwajczak, Investigation, Methodology | Aleksandra Szydłowska-Helbrych, Investigation, Methodology | Bernhard Kepplinger, Investigation, Writing – review and editing | Monika Pióro, Conceptualization, Investigation,

Supervision, Writing – review and editing | Dagmara Jakimowicz, Conceptualization, Formal analysis, Funding acquisition, Methodology, Project administration, Resources, Supervision, Validation, Visualization, Writing – original draft, Writing – review and editing

## ADDITIONAL FILES

The following material is available online.

### Supplemental Material

**Supplemental figures (Spectrum01752-23-s0001.pdf).** Fig. S1 to Fig. S7.
**Supplemental text (Spectrum01752-23-s0002.pdf).** Supplemental methods.
**Supplemental movie legends (Spectrum01752-23-s0003.pdf).** Legends for Movies S1 and S2.
**Table S1 (Spectrum01752-23-s0004.pdf).** Constructs used in the study.
**Table S2 (Spectrum01752-23-s0005.pdf).** Strains used in the study.
**Table S3 (Spectrum01752-23-s0006.pdf).** Oligonucleotides used in this study.
**Movie S1A (Spectrum01752-23-s0007.avi).** *papM* deletion alters EGFP-ParA fluorescence dynamics: control strain.
**Movie S1B (Spectrum01752-23-s0008.avi).** *papM* deletion alters EGFP-ParA fluorescence dynamics: *papM* background.
**Movie S2A (Spectrum01752-23-s0009.avi).** The mCherry-DivIVA redistribution during the cell cycle is affected by *parA* and *papM* deletion: wild type (WT) control strain.
**Movie S2B (Spectrum01752-23-s0010.avi).** The mCherry-DivIVA redistribution during the cell cycle is affected by *parA* and *papM* deletion: *parA* strain producing mCherry-DivIVA.
**Movie S2C (Spectrum01752-23-s0011.avi).** The mCherry-DivIVA redistribution during the cell cycle is affected by *parA* and *papM* deletion: *papM* strain producing mCherry-DivIVA.

### Open Peer Review

**PEER REVIEW HISTORY (review-history.pdf).** An accounting of the reviewer comments and feedback.

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
