## [Reviewer comments · Microbiology Spectrum]

Microbiology Spectrum

The interplay between the polar growth determinant DivIVA, the segregation protein ParA and their novel interaction partner PapM controls the *Mycobacterium smegmatis* cell cycle by modulation of DivIVA subcellular distribution

Izabela Matusiak, Agnieszka Strzałka, Patrycja Wadach, Martyna Gongerowska-Jac, Ewa Sz wajczak, Aleksandra Szydłowska-Helbrych, Bernhard Kepplinger, Monika Pióro, and Dagmara Jakimowicz

Corresponding Author(s): Dagmara Jakimowicz, Uniwersytet Wrocławski Wydział Biotechnologii

Review Timeline:

Submission Date:	April 26, 2023
Editorial Decision:	August 3, 2023
Revision Received:	September 28, 2023
Editorial Decision:	October 4, 2023
Revision Received:	October 5, 2023
Accepted:	October 6, 2023

Editor: Eric Cascales

Reviewer(s): Disclosure of reviewer identity is with reference to reviewer comments included in decision letter(s). The following individuals involved in review of your submission have agreed to reveal their identity: Takehiro Kado (Reviewer #1)

Transaction Report:

DOI: <https://doi.org/10.1128/spectrum.01752-23>

August 3, 2023

Prof. Dagmara Jakimowicz
Uniwersytet Wroclawski Wydział Biotechnologii
Department of Molecular Microbiology, Faculty of Biotechnology
Wroclaw
Poland

Re: Spectrum01752-23 (The interplay between the polar growth determinant DivIVA, the segregation protein ParA and their novel interaction partner PapM controls the Mycobacterium smegmatis cell cycle by modulation of DivIVA subcellular distribution)

Dear Dagmara,

Thank you for submitting your manuscript to Microbiology Spectrum. As we discussed recently, I encountered problems to identify suitable reviewers for your manuscript. One of the reviewers was not responsive anymore, and I have decided to proceed to the decision with only one reviewer. As you will see in the reviewer's comments, this reviewer appreciated your work and recommends publication once a few issues have been clarified. This reviewer is specifically concerned by the result of the DivIV phosphorylation, and suggests additional controls. She/he also requests more details on the biological replicates and statistical analyses. Based on the reviewer's opinion and recommendation, I encourage you to address the comments and invite you to submit a revised version of your work. When submitting the revised version of your paper, please provide (1) point-by-point responses to the issues raised by the reviewers as file type "Response to Reviewers," not in your cover letter, and (2) a PDF file that indicates the changes from the original submission (by highlighting or underlining the changes) as file type "Marked Up Manuscript - For Review Only". Please use this link to submit your revised manuscript - we strongly recommend that you submit your paper within the next 60 days or reach out to me. Detailed instructions on submitting your revised paper are below.

Link Not Available

Sincerely,
Eric

Eric Cascales

Journals Department
Reviewer comments:

Reviewer #1 (Comments for the Author):

Comments are uploaded as word file.

The authors focused on papM, which the authors newly found by using the bacterial two-hybrid (BTH) library of *M. smegmatis* genome. The interaction between PapM, ParA, and DivIVA were analyzed by BTH system, microscopic analysis in *E. coli*, and in vitro affinity chromatography. The function of PapM was further investigated in *M. smegmatis*, which originally expresses papM. The deletion of papM did not affect the cell growth but increased the immobile fraction of ParA and the establishment time of two daughter cell poles, while the overexpression extended the cell length when the authors used wild-type as the background. In addition, the authors showed some interesting results by deleting or overexpressing papM in parA deletion strains. This research is original and well written and provides a de novo binding partner of ParA. Overall, I agree with the idea that PapM interacts with ParA and DivIVA and co-relates to the cell segregation in *M. smegmatis*. However, I found some questionable arguments in the manuscripts. Please see the attached comments.

-Major points

1. The phosphorylation experiment of DivIVA needs further support. The authors expressed the kinase domain of PknB and PknA to phosphorylate DivIVA. The phosphorylation of DivIVA drastically decreased the co-localization of ParA in the presence of PapM (Fig 2F). The authors showed DivIVA was phosphorylated under the PknB overexpression state (Fig. S2), but the other random proteins of *E. coli* should be also phosphorylated by the over expression of PknB even if those proteins are from mycobacteria. Authors should make phosphomimic mutants of DivIVA to conclude "The presence of PapM enhances the dissociation of ParA from the DivIVA complex upon its phosphorylation (line 21-22), "ParA-DivIVA interaction is phosphorylation dependent" (Line 405), and "... its phosphorylation releases ParA" (line 424).
2. Biological replications are unclear. I could find the total number of the cells for the analysis but could not find how many biological replicates are done in the experiments. This is important information for readers, and a requirement for authors to show clear information about replications. In dot plots, authors can change the color of dots from the different biological replicates to make the graphs more informative [PMID: 32346721].
3. Statistical tests may not be appropriate in some figures. All statistical analyses have been done by student's t-test followed by the Holm method for multiple comparison. The t-test can be used if the datasets follow a gaussian distribution. In some of their data, I saw a non-gaussian distribution with my eyes (e.g., polarity of papM in PknB overexpression (Fig. 2H)). The authors must check the data distribution visually (with QQ-plots and histograms) or statistically (with tests such as D'Agostino-Pearson and Kolmogorov-Smirnov). And then, if the data do not follow gaussian distribution, authors have to use Mann-Whitney U test followed by Bonferroni correction for multiple comparison, or Kruskal-Wallis test followed by Dunn's multiple comparison.

-Minor points

1. Figure 1B. The anti-His western blotting is not a publication quality. Could the authors try once more? If this data is the representative data from biological replicates, authors should note this in the figure legend.
2. Figure S1C does not support the idea "...interaction interface may be located within the linker between DivIVA coiled-coil domains..." (Line 56-57) and "phosphorylation site Thr74 potentially may be involved in PapM binding." (Line 392-393). The figure shows the blue colony in which the author used T18C-DivIVA III-IV, which does not include the linker.
3. Figure 3C. This data should be shown as table format instead of figure.
4. Figure 3F. The annotations for images are missing.
5. Figure S5A. Why was the papM overexpression in ParA deletion mutant weaker than that of wild-type? I am wondering if the datasets came from technical triplicate in single biological replication.
6. Line 225, "parM" should be "papM".
7. Line 299-300, "This altered mobility could possibly result from modified ParA association with DNA which was earlier shown to predominantly affect ParA mobility". Does this mean that PapM affects the DNA binding affinity of ParA? I think this is not supported by data.
8. Line 326. NADA mainly visualizes peptidoglycan remodeled by L,D-transpeptidase instead of nascent peptide glycan [PMID: 30198841].
9. Line 388-389 "Elevated levels of PapM could interfere with ParA dynamics impairing segrosomes separation." is not a clear sentence. Does this mean that PapMs capture non-DNA-binding ParA to interfere with ParA dynamics?

Staff Comments:

Preparing Revision Guidelines

Please return the manuscript within 60 days; if you cannot complete the modification within this time period, please contact me. If you do not wish to modify the manuscript and prefer to submit it to another journal, please notify me of your decision immediately so that the manuscript may be formally withdrawn from consideration by Microbiology Spectrum.

The authors focused on *papM*, which the authors newly found by using the bacterial two-hybrid (BTH) library of *M. smegmatis* genome. The interaction between PapM, ParA, and DivIVA were analyzed by BTH system, microscopic analysis in *E.coli*, and *in vitro* affinity chromatography. The function of PapM was further investigated in *M. smegmatis*, which originally expresses *papM*. The deletion of *papM* did not affect the cell growth but increased the immobile fraction of ParA and the establishment time of two daughter cell poles, while the overexpression extended the cell length when the authors used wild-type as the background. In addition, the authors showed some interesting results by deleting or overexpressing *papM* in *parA* deletion strains. This research is original and well written and provides a *de novo* binding partner of ParA. Overall, I agree with the idea that PapM interacts with ParA and DivIVA and co-relates to the cell segregation in *M. smegmatis*. However, I found some questionable arguments in the manuscripts. Please see the attached comments.

-Major points

1. The phosphorylation experiment of DivIVA needs further support. The authors expressed the kinase domain of PknB and PknA to phosphorylate DivIVA. The phosphorylation of DivIVA drastically decreased the co-localization of ParA in the presence of PapM (Fig 2F). The authors showed DivIVA was phosphorylated under the PknB overexpression state (Fig. S2), but the other random proteins of *E.coli* should be also phosphorylated by the over expression of PknB even if those proteins are from mycobacteria. Authors should make phosphomimic mutants of DivIVA to conclude “The presence of PapM enhances the dissociation of ParA from the DivIVA complex upon its phosphorylation (line 21-22), “ParA-DivIVA interaction is phosphorylation dependent” (Line 405), and “... its phosphorylation releases ParA” (line 424).
2. Biological replications are unclear. I could find the total number of the cells for the analysis but could not find how many biological replicates are done in the experiments. This is important information for readers, and a requirement for authors to show clear information about replications. In dot plots, authors can change the color of dots from the different biological replicates to make the graphs more informative [PMID: 32346721].
3. Statistical tests may not be appropriate in some figures. All statistical analyses have been done by student's t-test followed by the Holm method for multiple comparison. The t-test can be used if the datasets follow a gaussian distribution. In some of their data, I saw a non-gaussian distribution with my eyes (e.g., polarity of *papM* in PknB overexpression (Fig. 2H)). The authors must check the data distribution visually (with QQ-plots and histograms) or statistically (with tests such as D'Agostino-Pearson and Kolmogorov-Smirnov). And then, if the data do not follow gaussian distribution, authors have to use Mann-Whitney U test followed by Bonferroni correction for multiple comparison, or Kruskal-Wallis test followed by Dunn's multiple comparison.

-Minor points

1. Figure 1B. The anti-His western blotting is not a publication quality. Could the authors try once more? If this data is the representative data from biological replicates, authors should note this in the figure legend.
2. Figure S1C does not support the idea "...interaction interface may be located within the linker between DivIVA coiled-coil domains..." (Line 56-57) and "phosphorylation site Thr74 potentially may be involved in PapM binding."(Line 392-393). The figure shows the blue colony in which the author used T18C-DivIVA III-IV, which does not include the linker.
3. Figure 3C. This data should be shown as table format instead of figure.
4. Figure 3F. The annotations for images are missing.
5. Figure S5A. Why was the *papM* overexpression in ParA deletion mutant weaker than that of wild-type? I am wondering if the datasets came from technical triplicate in single biological replication.
6. Line 225, "*parM*" should be "*papM*".
7. Line 299-300, "This altered mobility could possibly result from modified ParA association with DNA which was earlier shown to predominantly affect ParA mobility". Does this mean that PapM affects the DNA binding affinity of ParA? I think this is not supported by data.
8. Line 326. NADA mainly visualizes peptidoglycan remodeled by L,D-transpeptidase instead of nascent peptide glycan [PMID: 30198841].
9. Line 388-389 "Elevated levels of PapM could interfere with ParA dynamics impairing segrosomes separation." is not a clear sentence. Does this mean that PapMs capture non-DNA-binding ParA to interfere with ParA dynamics?

Response to Reviewers:

We would like to thank Reviewer for valuable comments. We have made suggested modifications, enhancing our manuscript impact and clarity. The detailed answers to comments are below.

Reviewer #1 (Comments for the Author):

The authors focused on papM, which the authors newly found by using the bacterial two-hybrid (BTH) library of *M. smegmatis* genome. The interaction between PapM, ParA, and DivIVA were analyzed by BTH system, microscopic analysis in *E. coli*, and in vitro affinity chromatography. The function of PapM was further investigated in *M. smegmatis*, which originally expresses papM. The deletion of papM did not affect the cell growth but increased the immobile fraction of ParA and the establishment time of two daughter cell poles, while the overexpression extended the cell length when the authors used wild-type as the background. In addition, the authors showed some interesting results by deleting or overexpressing papM in parA deletion strains. This research is original and well written and provides a de novo binding partner of ParA. Overall, I agree with the idea that PapM interacts with ParA and DivIVA and co-relates to the cell segregation in *M. smegmatis*. However, I found some questionable arguments in the manuscripts. Please see the attached comments.

-Major points

1. The phosphorylation experiment of DivIVA needs further support. The authors expressed the kinase domain of PknB and PknA to phosphorylate DivIVA. The phosphorylation of DivIVA drastically decreased the co-localization of ParA in the presence of PapM (Fig 2F). The authors showed DivIVA was phosphorylated under the PknB overexpression state (Fig. S2), but the other random proteins of *E. coli* should be also phosphorylated by the over expression of PknB even if those proteins are from mycobacteria. Authors should make phosphomimic mutants of DivIVA to conclude "The presence of PapM enhances the dissociation of ParA from the DivIVA complex upon its phosphorylation (line 21-22), "ParA-DivIVA interaction is phosphorylation dependent" (Line 405), and "... its phosphorylation releases ParA" (line 424).

We appreciate this very accurate comment, indeed the use of phosphoablative mutant of DivIVA is the best approach to confirm the significance of DivIVA phosphorylation. We have now constructed appropriate expression construct: pETDuet pknB_{KD} divIVAT74A and co-expressed these genes in presence of EGFP-ParA in *E. coli* BL21, finding that phosphoablative DivIV74A fully colocalised with EGFP-ParA in presence of His-PknB_{KD}. We have now included this result in the manuscript, as part of Fig. 2.

2. Biological replications are unclear. I could find the total number of the cells for the analysis but could not find how many biological replicates are done in the experiments. This is important information for readers, and a requirement for authors to show clear information about replications. In dot plots, authors can change the color of dots from the different biological replicates to make the graphs more informative [PMID: 32346721].

We have included information on biological replicates to the figures legends. All the data come from at least two biological replicates apart from the PALM experiment where data from multiple biological replicates were analysed, but images for final analyses were selected based on best signal

to-noise ratio. Unfortunately at this stage we cannot change the colour of data points, without performing the whole data analysis.

3. Statistical tests may not be appropriate in some figures. All statistical analyses have been done by student's t-test followed by the Holm method for multiple comparison. The t-test can be used if the datasets follow a gaussian distribution. In some of their data, I saw a non-gaussian distribution with my eyes (e.g., polarity of papM in PknB overexpression (Fig. 2H)). The authors must check the data distribution visually (with QQ-plots and histograms) or statistically (with tests such as D'Agostino-Pearson and Kolmogorov-Smirnov). And then, if the data do not follow gaussian distribution, authors have to use Mann-Whitney U test followed by Bonferroni correction for multiple comparison, or Kruskal-Wallis test followed by Dunn's multiple comparison.

Indeed, approximately normal distribution of samples is listed as a requirement for the T-student test. However, given a sufficient sample size, this does not have to be the case. According to the Central Limit Theorem (CLT), repeated sampling from a non-normal population will produce a normal distribution of sampling means. This Normal distribution of an average underlies the validity of the T-student test. Non-parametric test such as Wilcoxon test are the most useful, when the sample size is small and can be misleading when used for large datasets. (Lumley et al, 2002, Fagerland, 2012)

We repeated the statistical analysis for data presented on figures 2 and 5, where sample's distributions were skewed and did not follow normal distribution using Wilcoxon rank-sum test (Mann–Whitney U test) instead of T-student test, with the same method for multiple comparisons correction (Holm). This analyses produced similar p-values with the exception of Figure 2H right panel and Fig. 5B. Therefore we exchanged those figures. We show below the comparison of p values obtained with both statistical approaches.

Figure	Panel	Comparison	T-student test p-value	Wilcoxon test p-value
2	H, left	ParA+DivIVA vs ParA + DivIVA + PapM	0.7330	0.5279
2	H, left	ParA+DivIVA vs ParA+DivIVA + PknB	3.6e-07	3.3e-08
2	H, left	ParA+DivIVA vs ParA+DivIVA + PapM + PknB	< 2e-16	< 2e-16
2	H, left	ParA+DivIVA + PknB vs ParA+DivIVA + PapM + PknB	< 2e-16	< 2e-16
2	H, right	DivIVA + PapM vs ParA + DivIVA + PapM	0.0082	0.2900
2	H, right	DivIVA + PapM vs ParA + DivIVA + PapM + PknB	6.0e-14	1.6e-08
2	H, right	DivIVA + PapM vs DivIVA + PapM + PknB	1.9e-09	4.4e-09
2	H, right	ParA + DivIVA + PapM vs ParA + DivIVA + PapM + PknB	0.0082	0.2300
2	H, right	ParA + DivIVA + PapM + PknB vs DivIVA + PapM + PknB	0.0230	0.2900
5	A	WT vs Δ papM	0.2000	0.0900
5	A	WT vs Δ parA	0.0120	0.0190
5	B	WT vs Δ papM	0.1300	0.0620
5	B	WT vs Δ parA	0.0270	0.1500

5	C	WT vs Δ papM	0.3554	0.9747
5	C	WT vs Δ parA	0.0012	0.0006

-Minor points

1. Figure 1B. The anti-His western blotting is not a publication quality. Could the authors try once more? If this data is the representative data from biological replicates, authors should note this in the figure legend.

The results presented in figure 1 are representative of 4 independent experimental approaches which each time deliver the same clear result (while Western presented in figure1 was the best technical replicate) - we have now included this information in the figure legend.

2. Figure S1C does not support the idea "...interaction interface may be located within the linker between DivIVA coiled-coil domains..." (Line 56-57) and "phosphorylation site Thr74 potentially may be involved in PapM binding."(Line 392-393). The figure shows the blue colony in which the author used T18C-DivIVA III-IV, which does not include the linker.

We have now included the detailed information on the fragments used in BTH assay, which shows that T18C-DivIVA III-IV (68-143 aa) includes the phosphorylated Thr74. To be more precise, we have also rephrased the text in the result as follows: "interaction interface may encompass the linker between DivIVA coiled-coil domains and fragment of second coiled-coil..."

3. Figure 3C. This data should be shown as table format instead of figure.

OK, Fig 3C is now Table 1

4. Figure 3F. The annotations for images are missing.

Corrected, this figure is now Fig. 3A inset.

5. Figure S5A. Why was the papM overexpression in ParA deletion mutant weaker than that of wild-type? I am wondering if the datasets came from technical triplicate in single biological replication.

The dataset are from 4 biological replicates (we added this information into the figure legend), but we cannot explain why papM transcript levels are lower in absence of ParA. However, taking into account the high variation of papM transcript levels in wild type the difference between wild type and Δ parA mutant may be not significant.

6. Line 225, "parM" should be "papM".

corrected

7. Line 299-300, "This altered mobility could possibly result from modified ParA association with DNA which was earlier shown to predominantly affect ParA mobility". Does this

mean that PapM affects the DNA binding affinity of ParA? I think this is not supported by data.

We believe that suggested by PALM experiment change in the ParA mobility may be due to its altered DNA binding. Our earlier work showed that alterations of ParA-DNA binding have a large impact on ParA mobility. We have added the appropriate reference here.

8. Line 326. NADA mainly visualizes peptidoglycan remodeled by L,D-transpeptidase instead of nascent peptide glycan [PMID: 30198841].

Corrected (deleted “nascent”).

9. Line 388-389 "Elevated levels of PapM could interfere with ParA dynamics impairing segrosomes separation." is not a clear sentence. Does this mean that PapMs capture non-DNA-binding ParA to interfere with ParA dynamics?

That is one of the possibilities that should be considered. We have rephrased the text as follows: “Elevated levels of PapM could interfere with ParA-DNA and/or ParA-DivIVA interaction affecting ParA dynamics and impairing segrosomes separation.”

October 4, 2023

Prof. Dagmara Jakimowicz
Uniwersytet Wrocławski Wydział Biotechnologii
Department of Molecular Microbiology, Faculty of Biotechnology
Wrocław
Poland

Re: Spectrum01752-23R1 (The interplay between the polar growth determinant DivIVA, the segregation protein ParA and their novel interaction partner PapM controls the *Mycobacterium smegmatis* cell cycle by modulation of DivIVA subcellular distribution)

Dear Prof. Dagmara Jakimowicz:

Thank you for submitting your revised manuscript, which has been sent to the original reviewer. As you will see, this reviewer acknowledges that you properly addressed all the questions raised in the first round of review. One typo has however been noticed. I therefore invite you to make this last change and to submit your revised manuscript. I will then proceed to final acceptance.

When submitting the revised version of your paper, please provide (1) point-by-point responses to the issues raised by the reviewers as file type "Response to Reviewers," not in your cover letter, and (2) a PDF file that indicates the changes from the original submission (by highlighting or underlining the changes) as file type "Marked Up Manuscript - For Review Only". Please use this link to submit your revised manuscript. Detailed instructions on submitting your revised paper are below.

Link Not Available

Sincerely,

Eric Cascales

Reviewer comments:

Reviewer #1 (Comments for the Author):

The authors have done a great job in revising the manuscript and addressing the issues raised by the reviewer. Finding PapM as a new modulator of DivIVA-parA in *M.smegmatis* is a novel, interesting and intriguing.

The authors have addressed all my questions.

One minor comment- Right panels of figure 4 say "papA" and "parM". I believe those should be "parA" and "papM"

Preparing Revision Guidelines

Please return the manuscript within 60 days; if you cannot complete the modification within this time period, please contact me. If you do not wish to modify the manuscript and prefer to submit it to another journal, please notify me of your decision immediately so that the manuscript may be formally withdrawn from consideration by Microbiology Spectrum.

Response to Reviewers:

We would like to thank Reviewer for valuable comments. We have made suggested modifications, enhancing our manuscript impact and clarity. The detailed answers to comments are below.

Reviewer #1 (Comments for the Author):

The authors focused on papM, which the authors newly found by using the bacterial two-hybrid (BTH) library of *M. smegmatis* genome. The interaction between PapM, ParA, and DivIVA were analyzed by BTH system, microscopic analysis in *E. coli*, and in vitro affinity chromatography. The function of PapM was further investigated in *M. smegmatis*, which originally expresses papM. The deletion of papM did not affect the cell growth but increased the immobile fraction of ParA and the establishment time of two daughter cell poles, while the overexpression extended the cell length when the authors used wild-type as the background. In addition, the authors showed some interesting results by deleting or overexpressing papM in parA deletion strains. This research is original and well written and provides a de novo binding partner of ParA. Overall, I agree with the idea that PapM interacts with ParA and DivIVA and co-relates to the cell segregation in *M. smegmatis*. However, I found some questionable arguments in the manuscripts. Please see the attached comments.

-Major points

1. The phosphorylation experiment of DivIVA needs further support. The authors expressed the kinase domain of PknB and PknA to phosphorylate DivIVA. The phosphorylation of DivIVA drastically decreased the co-localization of ParA in the presence of PapM (Fig 2F). The authors showed DivIVA was phosphorylated under the PknB overexpression state (Fig. S2), but the other random proteins of *E. coli* should be also phosphorylated by the over expression of PknB even if those proteins are from mycobacteria. Authors should make phosphomimic mutants of DivIVA to conclude "The presence of PapM enhances the dissociation of ParA from the DivIVA complex upon its phosphorylation (line 21-22), "ParA-DivIVA interaction is phosphorylation dependent" (Line 405), and "... its phosphorylation releases ParA" (line 424).

We appreciate this very accurate comment, indeed the use of phosphoablative mutant of DivIVA is the best approach to confirm the significance of DivIVA phosphorylation. We have now constructed appropriate expression construct: pETDuet pknB_{KD} divIVAT74A and co-expressed these genes in presence of EGFP-ParA in *E. coli* BL21, finding that phosphoablative DivIV74A fully colocalised with EGFP-ParA in presence of His-PknB_{KD}. We have now included this result in the manuscript, as part of Fig. 2.

2. Biological replications are unclear. I could find the total number of the cells for the analysis but could not find how many biological replicates are done in the experiments. This is important information for readers, and a requirement for authors to show clear information about replications. In dot plots, authors can change the color of dots from the different biological replicates to make the graphs more informative [PMID: 32346721].

We have included information on biological replicates to the figures legends. All the data come from at least two biological replicates apart from the PALM experiment where data from multiple biological replicates were analysed, but images for final analyses were selected based on best signal

to-noise ratio. Unfortunately at this stage we cannot change the colour of data points, without performing the whole data analysis.

3. Statistical tests may not be appropriate in some figures. All statistical analyses have been done by student's t-test followed by the Holm method for multiple comparison. The t-test can be used if the datasets follow a gaussian distribution. In some of their data, I saw a non-gaussian distribution with my eyes (e.g., polarity of papM in PknB overexpression (Fig. 2H)). The authors must check the data distribution visually (with QQ-plots and histograms) or statistically (with tests such as D'Agostino-Pearson and Kolmogorov-Smirnov). And then, if the data do not follow gaussian distribution, authors have to use Mann-Whitney U test followed by Bonferroni correction for multiple comparison, or Kruskal-Wallis test followed by Dunn's multiple comparison.

Indeed, approximately normal distribution of samples is listed as a requirement for the T-student test. However, given a sufficient sample size, this does not have to be the case. According to the Central Limit Theorem (CLT), repeated sampling from a non-normal population will produce a normal distribution of sampling means. This Normal distribution of an average underlies the validity of the T-student test. Non-parametric test such as Wilcoxon test are the most useful, when the sample size is small and can be misleading when used for large datasets. (Lumley et al, 2002, Fagerland, 2012)

We repeated the statistical analysis for data presented on figures 2 and 5, where sample's distributions were skewed and did not follow normal distribution using Wilcoxon rank-sum test (Mann–Whitney U test) instead of T-student test, with the same method for multiple comparisons correction (Holm). This analyses produced similar p-values with the exception of Figure 2H right panel and Fig. 5B. Therefore we exchanged those figures. We show below the comparison of p values obtained with both statistical approaches.

Figure	Panel	Comparison	T-student test p-value	Wilcoxon test p-value
2	H, left	ParA+DivIVA vs ParA + DivIVA + PapM	0.7330	0.5279
2	H, left	ParA+DivIVA vs ParA+DivIVA + PknB	3.6e-07	3.3e-08
2	H, left	ParA+DivIVA vs ParA+DivIVA + PapM + PknB	< 2e-16	< 2e-16
2	H, left	ParA+DivIVA + PknB vs ParA+DivIVA + PapM + PknB	< 2e-16	< 2e-16
2	H, right	DivIVA + PapM vs ParA + DivIVA + PapM	0.0082	0.2900
2	H, right	DivIVA + PapM vs ParA + DivIVA + PapM + PknB	6.0e-14	1.6e-08
2	H, right	DivIVA + PapM vs DivIVA + PapM + PknB	1.9e-09	4.4e-09
2	H, right	ParA + DivIVA + PapM vs ParA + DivIVA + PapM + PknB	0.0082	0.2300
2	H, right	ParA + DivIVA + PapM + PknB vs DivIVA + PapM + PknB	0.0230	0.2900
5	A	WT vs Δ papM	0.2000	0.0900
5	A	WT vs Δ parA	0.0120	0.0190
5	B	WT vs Δ papM	0.1300	0.0620
5	B	WT vs Δ parA	0.0270	0.1500

5	C	WT vs Δ papM	0.3554	0.9747
5	C	WT vs Δ parA	0.0012	0.0006

-Minor points

1. Figure 1B. The anti-His western blotting is not a publication quality. Could the authors try once more? If this data is the representative data from biological replicates, authors should note this in the figure legend.

The results presented in figure 1 are representative of 4 independent experimental approaches which each time deliver the same clear result (while Western presented in figure1 was the best technical replicate) - we have now included this information in the figure legend.

2. Figure S1C does not support the idea "...interaction interface may be located within the linker between DivIVA coiled-coil domains..." (Line 56-57) and "phosphorylation site Thr74 potentially may be involved in PapM binding."(Line 392-393). The figure shows the blue colony in which the author used T18C-DivIVA III-IV, which does not include the linker.

We have now included the detailed information on the fragments used in BTH assay, which shows that T18C-DivIVA III-IV (68-143 aa) includes the phosphorylated Thr74. To be more precise, we have also rephrased the text in the result as follows: "interaction interface may encompass the linker between DivIVA coiled-coil domains and fragment of second coiled-coil..."

3. Figure 3C. This data should be shown as table format instead of figure.

OK, Fig 3C is now Table 1

4. Figure 3F. The annotations for images are missing.

Corrected, this figure is now Fig. 3A inset.

5. Figure S5A. Why was the papM overexpression in ParA deletion mutant weaker than that of wild-type? I am wondering if the datasets came from technical triplicate in single biological replication.

The dataset are from 4 biological replicates (we added this information into the figure legend), but we cannot explain why papM transcript levels are lower in absence of ParA. However, taking into account the high variation of *papM* transcript levels in wild type the difference between wild type and Δ parA mutant may be not significant.

6. Line 225, "parM" should be "papM".

corrected

7. Line 299-300, "This altered mobility could possibly result from modified ParA association with DNA which was earlier shown to predominantly affect ParA mobility". Does this

mean that PapM affects the DNA binding affinity of ParA? I think this is not supported by data.

We believe that suggested by PALM experiment change in the ParA mobility may be due to its altered DNA binding. Our earlier work showed that alterations of ParA-DNA binding have a large impact on ParA mobility. We have added the appropriate reference here.

8. Line 326. NADA mainly visualizes peptidoglycan remodeled by L,D-transpeptidase instead of nascent peptide glycan [PMID: 30198841].

Corrected (deleted “nascent”).

9. Line 388-389 "Elevated levels of PapM could interfere with ParA dynamics impairing segrosomes separation." is not a clear sentence. Does this mean that PapMs capture non-DNA-binding ParA to interfere with ParA dynamics?

That is one of the possibilities that should be considered. We have rephrased the text as follows: “Elevated levels of PapM could interfere with ParA-DNA and/or ParA-DivIVA interaction affecting ParA dynamics and impairing segrosomes separation.”

October 6, 2023

Prof. Dagmara Jakimowicz
Uniwersytet Wrocławski Wydział Biotechnologii
Department of Molecular Microbiology, Faculty of Biotechnology
Wrocław
Poland

Re: Spectrum01752-23R2 (The interplay between the polar growth determinant DivIVA, the segregation protein ParA and their novel interaction partner PapM controls the Mycobacterium smegmatis cell cycle by modulation of DivIVA subcellular distribution)

Dear Dagmara,

Thank you for incorporating the last change in your manuscript. I am pleased to accept your manuscript in Microbiol Spectrum, and I am forwarding it to the ASM Journals Department for publication. You will be notified when your proofs are ready to be viewed.

Sincerely,
Eric

Eric Cascales
Editor, Microbiology Spectrum
